# B1 SINE-binding ZFP266 impedes mouse iPSC generation through suppression of chromatin opening mediated by reprogramming factors

Daniel F. Kaemena[1], Masahito Yoshihara [2,3,4], Meryam Beniazza[1], James Ashmore[1], Suling Zhao[1], Mårten Bertenstam[5], Victor Olariu[5], Shintaro Katayama [2,6,7], Keisuke Okita [8], Simon R. Tomlinson[1], Kosuke Yusa [9,10] & Keisuke Kaji [1]✉

Induced pluripotent stem cell (iPSC) reprogramming is inefficient and understanding the molecular mechanisms underlying this inefficiency holds the key to successfully control cellular identity. Here, we report 24 reprogramming roadblock genes identified by CRISPR/Cas9-mediated genome-wide knockout (KO) screening. Of these, depletion of the predicted KRAB zinc finger protein (KRAB-ZFP) *Zfp266* strongly and consistently enhances murine iPSC generation in several reprogramming settings, emerging as the most robust roadblock. We show that ZFP266 binds Short Interspersed Nuclear Elements (SINEs) adjacent to binding sites of pioneering factors, OCT4 (POU5F1), SOX2, and KLF4, and impedes chromatin opening. Replacing the KRAB co-suppressor with co-activator domains converts ZFP266 from an inhibitor to a potent facilitator of iPSC reprogramming. We propose that the SINE-KRAB-ZFP interaction is a critical regulator of chromatin accessibility at regulatory elements required for efficient cellular identity changes. In addition, this work serves as a resource to further illuminate molecular mechanisms hindering reprogramming.

The reprogramming of somatic cells into iPSCs via the overexpression of *Oct4 (Pou5f1)*, *Sox2*, *Klf4*, and *c-Myc* (OSKM) has provided an important tool for medical research and cell therapies[1]. Equally importantly, the generation of fully functional iPSCs that are indistinguishable from ESCs from somatic cells has demonstrated that cellular identity can be completely converted from one type to another by overexpression of master transcription factors. This has provided a model system to understand how to control cellular identity. Inhibition of *Trp53* and *Cdkn1a* (*p21*) revealed OSKM-induced apoptosis and senescence as major roadblock of iPSC generation[2-8]. Knockdown of *Dot1l* and *Suv39h1* has demonstrated H3K79me and H3K9me3 as critical epigenetic modifications that impede this cell conversion[9,10].

[1]Centre for Regenerative Medicine, Institute for Regeneration and Repair, University of Edinburgh, Edinburgh BioQuarter, 5 Little France Drive, Edinburgh, Scotland, UK. [2]Department of Biosciences and Nutrition, Karolinska Institutet, Stockholm, Sweden. [3]Institute for Advanced Academic Research, Chiba University, Chiba, Japan. [4]Department of Artificial Intelligence Medicine, Graduate School of Medicine, Chiba University, Chiba, Japan. [5]Computational Biology and Biological Physics, Lund University, Lund, Sweden. [6]Research Programs Unit, Stem Cells and Metabolism Research Program (STEMM), University of Helsinki, Helsinki, Finland. [7]Folkhälsan Research Center, Helsinki, Finland. [8]Center for iPS Cell Research and Application, Kyoto University, Kyoto, Japan. [9]Stem Cell Genetics, Wellcome Sanger Institute, Hinxton, Cambridge, UK. [10]Stem Cell Genetics, Institute for Life and Medical Sciences, Kyoto University, Kyoto, Japan. ✉e-mail: keisuke.kaji@ed.ac.uk

Thus, identifying genes that act against successful reprogramming provides a foundation to understand critical molecular mechanisms involved in pluripotency induction.

Transposable elements (TEs), which constitute approximately 45% of mouse and human genomes, can take part in gene expression regulation as cis-regulatory elements or non-coding RNAs[11]. Long terminal repeat (LTR) retrotransposons, long and short interspersed nuclear elements (LINEs and SINEs) are the three major classes of human/mouse TEs and the functional importance of the first two groups in pluripotent cells has been described[12,13]. Knockdown of the long interspersed nuclear element 1 (LINE1) inhibits mouse ESC self-renewal and induces a transition to a 2-cell-like state[12]. KLF4 activates transcription of LTR retrotransposon human endogenous retrovirus subfamily H (HERVH) during reprogramming, the down-regulation of which is critical for exit from the pluripotent state of human iPSCs[13]. Chromatin accessibility at SINEs, which constitute ~25% of TEs, is particularly high in mouse pre-implantation embryos and ESCs[14], but the functional importance of this has not yet been demonstrated. Krüppel-associated box (KRAB) zinc-finger proteins (ZFPs) form the largest TF family in mouse and human genomes with over 300 members[15]. They have evolved to suppress the expression and transposition of rapidly mutating TEs, with about two-thirds of human KRAB-ZFPs estimated to bind to TEs[16]. Thus, some KRAB-ZFPs might be involved in the regulation of the above-mentioned pluripotency-associated LINE1 and HERVH expression. The binding of KRAB-ZFPs on TEs can also regulate the expression of nearby genes[17]. Knockout of the KRAB-ZFP cluster in chromosome 2 or chromosome 4, which contains 40 or 21 KRAB-ZFPs respectively, in mouse ESCs preferentially up-regulated genes near specific classes of LTR retrotransposons and LINEs[18]. Overexpression of ZNF611 in human ESCs down-regulated genes near primate-specific SINE-VNTR-Alu (SVA) retrotransposons[19]. Nevertheless, only a small number of KRAB-ZFPs that predominantly bind SINEs have been reported[16,18], and the importance of KRAB-ZFP/SINE interaction for gene expression regulation is not well understood.

Here, we report an unbiased genome-wide CRISPR KO screen with a library containing 90,230 sgRNAs targeting 18,424 protein-coding genes focusing on iPSC reprogramming. This screen identifies 24 reprogramming roadblock genes, including previously reported 8. Of those, KO of the previously uncharacterized mouse KRAB-ZFP gene *Zfp266* accelerates the kinetics of reprogramming and improved the efficiency of iPSC generation by 4- to 10-fold in various reprogramming contexts. In this work, we reveal that ZFP266 is able to bind to B1 SINEs adjacent to OSK binding sites during reprogramming, where it impedes chromatin opening. Thus, these B1 SINEs containing loci are critical genetic elements that modulate the efficiency of OSKM-mediated mouse iPSC generation. This work serves as a resource for better understanding reprogramming mechanisms and highlights SINEs as a previously undescribed TE class involved in pluripotency induction.

## Results

### CRISPR/Cas9-mediated genome-wide KO screening identified 24 reprogramming roadblock genes

We have previously generated a Cas9 expressing mouse ES cell line, named Cas9 TNG MKOS, with a *Nanog-GFP-ires-Puro* reporter and a doxycycline-inducible *MKOS-ires-mOrange* polycistronic reprogramming cassette in the Sp3 locus (Supplementary Fig. 1A, B)[20,21]. Efficient KO by lentiviral sgRNA delivery in both Cas9 TNG MKOS ESCs and mouse embryonic fibroblasts (MEFs), generated through morula aggregation of these ESCs, was confirmed using sgRNAs against *Icam1* and *Cd44*, respectively, resulting in >80% loss of protein within 72 h (Supplementary Fig. 1C, D). Reprogramming of Cas9 TNG MKOS MEFs following sgRNA transduction against known roadblock genes *Trp53* and *Rb1*[3–8,22], and essential genes *Pou5f1* and *Kdm6a*[23], reproduced the expected reprogramming enhancement and reduction phenotypes

(Supplementary Fig. 1E–G), confirming that the CRISPR-based KO system is a powerful tool to investigate gene function in reprogramming. We then performed genome-wide KO screening using a previously published lentiviral sgRNA library[24], with an optimized reprogramming condition consisting of 8 days of reprogramming factor expression followed by 8 days of puromycin selection for *Nanog*-GFP+ iPSCs (Fig. 1A and Supplementary Fig. 1H–K). This condition resulted in an average coverage of ~170 MEFs/sgRNA/screening replicate. Genomic DNA from flow-sorted *Nanog*-GFP+ iPSCs was then collected in triplicate, and integrated sgRNAs were Illumina-sequenced after PCR amplification alongside the original sgRNA plasmid library.

The normalized read counts of all sgRNAs and analysis of the screening results with MAGeCK[25] are available in Supplementary Data 1, 2 and at https://kaji-crispr-screen-updated.netlify.app. Using a false discovery rate (FDR) < 0.1 as a cut-off, we identified 24 genes as reprogramming roadblocks (Fig. 1B). This included 5 previously characterized genes: *Trp53, Cdkn1a, Dot1l,* and AP-1 transcription factor members *Jun* and *Fosl2*[9,26], and 3 genes previously uncharacterized yet identified in other screens, *Men1, Gtf2i, and Cdk13*[27,28], signifying the robustness of our screen (Fig. 1C, D). When the top 3 ranked sgRNAs for each gene were individually tested, transduction of *Trp53* and *Cdkn1a* sgRNAs produced the largest increase in *Nanog*-GFP+ colony numbers (Fig. 1E, F), although they also significantly increased the number of partially reprogrammed colonies (Fig. 1F and Supplementary Fig. 2A). Transduction of sgRNAs targeting all other genes, except for *Cdk13*, enhanced *Nanog*-GFP+ colony formation between 2- and 6-fold (Fig. 1E and Supplementary Fig. 2A), verifying the inhibitory effects of the reprogramming roadblock genes. Expression of the validated 23 roadblock genes during reprogramming did not follow any common particular pattern and many of them exhibited consistently low expression, compared to the common housekeeping genes or the reprogramming and pluripotency marker genes[29] (Supplementary Fig. 2B). This highlights the advantage of functional screening to identify their inhibitory effects over relying on expression profiling.

Of the roadblock genes we identified, KO of *Fam1222a, Zfp266, Bcorl1, Usp28, Usp34, Zc3h10, Scaf8,* and *Spop* resulted in a > 4-fold enhancement similar to or better than previously reported roadblocks (Fig. 1E). We therefore further characterized these 8 reprogramming roadblocks (Fig. 1F, blue), alongside the previously reported roadblocks *Trp53, Cdkn1a, Men1, Dot1l* and *Gtf2i* (Fig. 1F, green), as 13 top roadblocks.

### *Zfp266* KO consistently enhances and accelerates the attainment of pluripotency

Reprogramming roadblock function is influenced by multiple elements such as the stoichiometry or expression levels of OSKM, culture conditions, and starting cell types[20]. Thus, we examined the KO effects of our 13 top roadblocks in different reprogramming contexts. We first performed *piggyBac* transposon-based reprogramming with an *MKOS* or STEMCCA (*OKSM*) reprogramming cassette[30,31] (Supplementary Fig. 3A). The STEMCCA cassette expresses lower levels of KLF4 protein due to an N-terminal truncation following a 2A peptide[32], resulting in inefficient mesenchymal-epithelial transition (MET) and a higher proportion of partially reprogrammed cells[20,33]. *piggyBac* delivery of the MKOS cassette together with each sgRNA against all the 13 roadblocks enhanced reprogramming of Cas9 Nanog-GFP MEFs as seen with Cas9 TNG MKOS MEFs before (Fig. 2A, B), despite a markedly lower KO efficiency with the *piggyBac* system compared to lentiviral sgRNA delivery (Supplementary Fig. 3B). In STEMCCA-mediated *piggyBac* reprogramming, KO of all roadblocks, except *Cdkn1a, Fam122a,* and *Zc3h10*, increased the number of *Nanog*-GFP+ colonies, although *Cdkn1a* and *Fam122a* KO also drastically increased *Nanog*-GFP- colony numbers (Fig. 2C, D). It is likely that *Fam122a* KO suppresses reprogramming-induced senescence/apoptosis like *Cdkn1a* KO, and high exogenous KLF4 expression is required to push those partially reprogrammed cells toward an iPSC state. In particular,

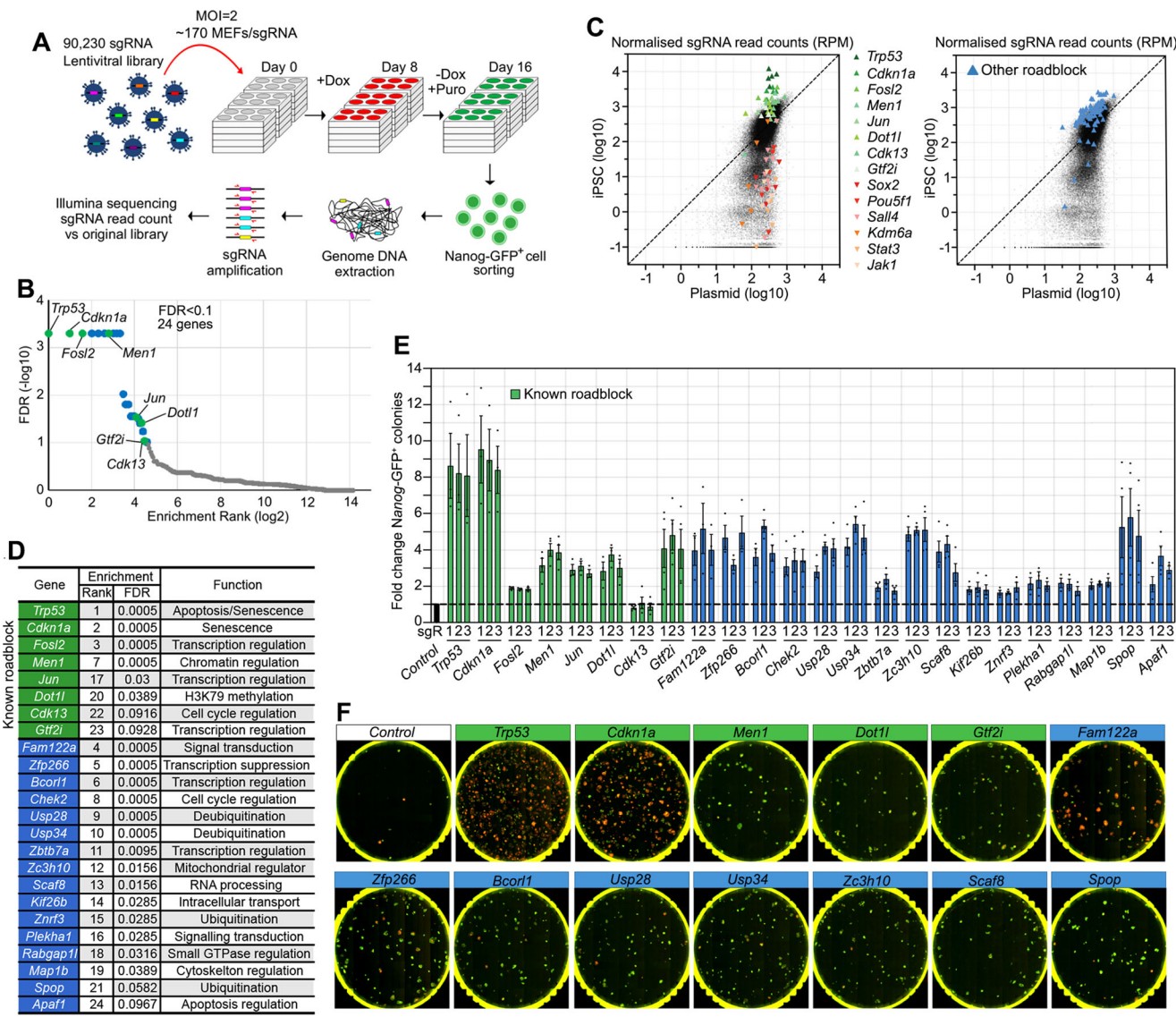

**Fig. 1 | A genome-wide CRISPR screen identifies reprogramming roadblock genes. A** Schematic diagram of the screening strategy. sgRNA library infected Cas9 TNG MKOS MEFs were cultured in +dox for 8 days then in -dox and +Puro for additional 8 days. Integrated sgRNAs were amplified from *Nanog*-GFP⁺ cells for Illumina sequencing. **B** Enrichment FDR ranking with MAGeCK. 24 genes, including eight previously reported (green) roadblock genes, were identified using a cut-off of FDR < 0.1. **C** Normalized sgRNA read counts in the initial plasmid library versus mutant iPSC pool. sgRNAs against previously reported roadblock genes (green) and genes essential for reprogramming (red/orange) exhibited expected enrichment/ depletion respectively (left). sgRNAs against other 16 roadblock genes identified in this screen are highlighted in blue (right). **D** Enrichment rank, FDR, and function of the 24 discovered roadblock genes. **E** Validation of the screen result with three individual sgRNAs per gene. This graph is a summary of five data sets shown in Supplementary Figure 2A. **F** Representative whole-well images of KO reprogramming of 13 top roadblocks from (**E**). Previously reported roadblock genes were labeled in green. Red; mOrange, Green; *Nanog*-GFP. Source data are provided as a Source Data file.

the KO of *Zfp266* increased numbers of *Nanog*-GFP⁺ colonies ~10-fold with almost all colonies expressing *Nanog*-GFP unlike *Cdkn1a* and *Fam122a* KO (Fig. 2C, D). When *piggyBac MKOS* + sgRNA vectors were used to reprogram *Cas9* expressing neural stem cells (NSCs), sgRNAs against *Men1, Fam122a, Zfp266* and *Usp34* increased reprogramming efficiency, with KO of *Zfp266* again leading to the greatest enhancement in NANOG⁺ colony formation (~5-fold) (Fig. 2E, F). When we explored reprogramming kinetics by assessing expression changes of reprogramming markers, CD44, ICAM1, and *Nanog*-GFP[29] using Cas9 TNG MKOS MEFs, KO of 5 genes, *Men1, Gtf2i, Dot1l, Zfp266,* and *Zc3h10,* demonstrated accelerated reprogramming (Fig. 2G, H, and Supplementary Fig. 3C). In summary, KO of *Zfp266* exhibited the most context-independent and robust reprogramming enhancement amongst all roadblock genes we identified. Furthermore, it could also increase the number of NANOG⁺ iPSC colonies and decrease the number of NANOG⁻

iPSC colonies in the presence of *Trp53, Cdkn2a,* or *Fam122a* gRNAs, all of which tend to generate a high number of NANOG⁻ partially reprogrammed colonies, indicating that reprogramming enhancement by *Zfp266* KO is not due to circumvention of reprogramming factor-induced senescence and apoptosis (Supplementary Figure 3D). We, therefore, investigated further how ZFP266 impedes the reprogramming process.

## ZFP266 impedes activation of pluripotency genes via its KRAB domain

*Zfp266* is predicted to encode a KRAB-ZF protein with a singular KRAB-A module in the N-terminus and putative DNA binding domain with 12x C2H2 type zinc finger array in the C-terminus (Fig. 3A). KRAB domains are known to interact with co-suppressor KAP-1/TRIM28, a scaffold protein that can recruit epigenetic modifiers and promote the

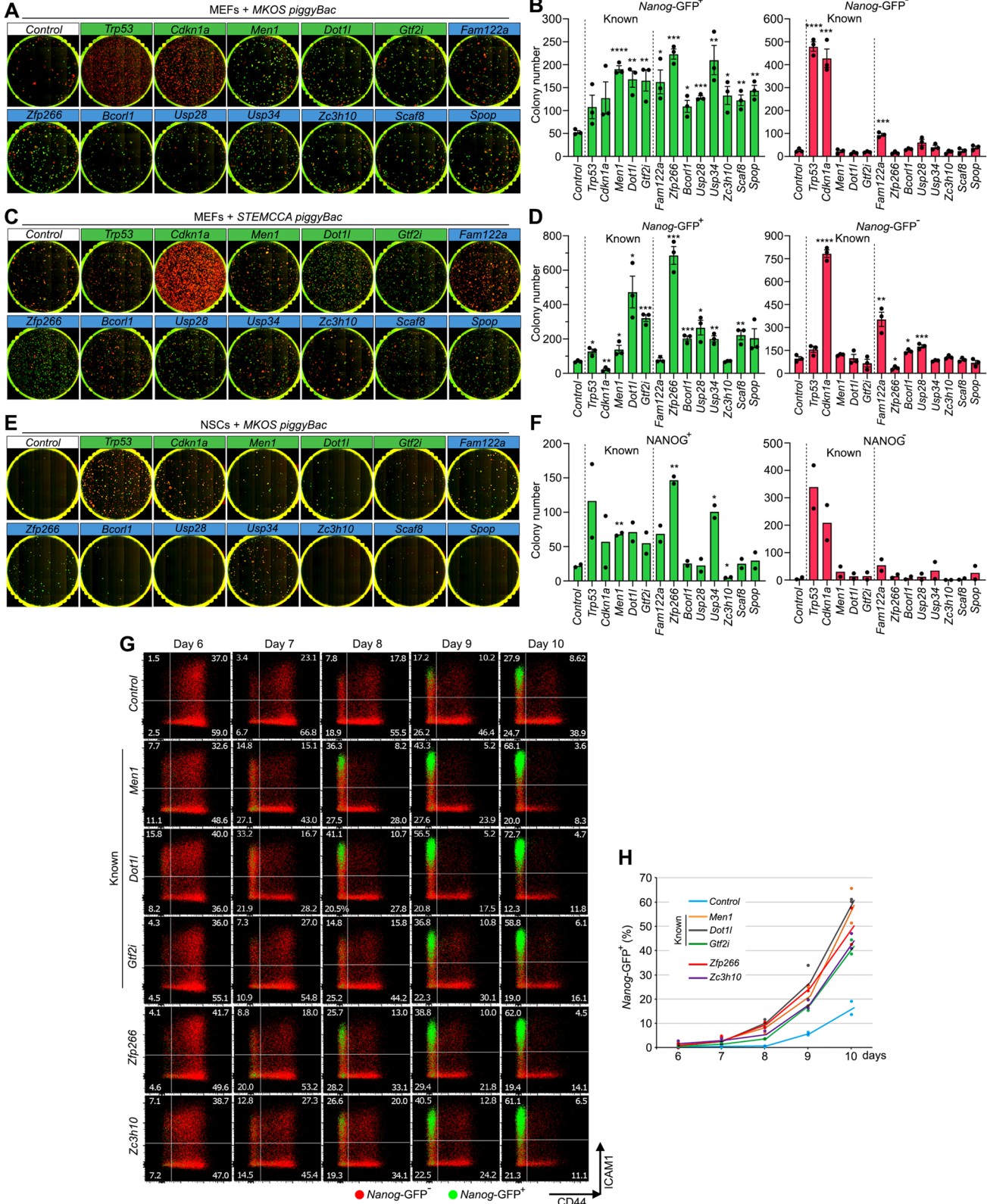

**Fig. 2 | Characterization of the roadblock gene KO in different reprogramming systems and kinetics. A**, **C** *Cas9* expressing *Nanog*-GFP MEF reprogramming with MKOS (**A**), STEMCCA (**C**) *piggyBac* transposons with sgRNA expression at day 15. Red; mOrange, Green; *Nanog*-GFP. **B**, **D** *Nanog*-GFP⁺ and *Nanog*-GFP⁻ mean colony numbers from **A** and **C**. The graphs in **B** and **D** represent an average of three independent experiments. **E** *Cas9* expressing NSC reprogramming with MKOS *piggyBac* transposons with sgRNA expression at day 15. Green; immuno-fluorescence for NANOG. **F** NANOG + and NANOG- mean colony numbers from **E**.

The graph represents an average of two independent experiments. **G** Accelerated CD44/ICAM/*Nanog*-GFP expression changes by sgRNA expression against the roadblock genes (*n* = 2). Red; *Nanog*-GFP- cells, Green; *Nanog*-GFP + cells. **H** Quantification of *Nanog*-GFP + cells from day 6 to 10 of reprogramming. The graph represents an average of two independent experiments. In **B**, **D**, **F**, ****$p < 0.0001$, ***$p < 0.001$, **$p < 0.01$, *$p < 0.05$. Source data and exact *p*-values are provided as a Source Data file.

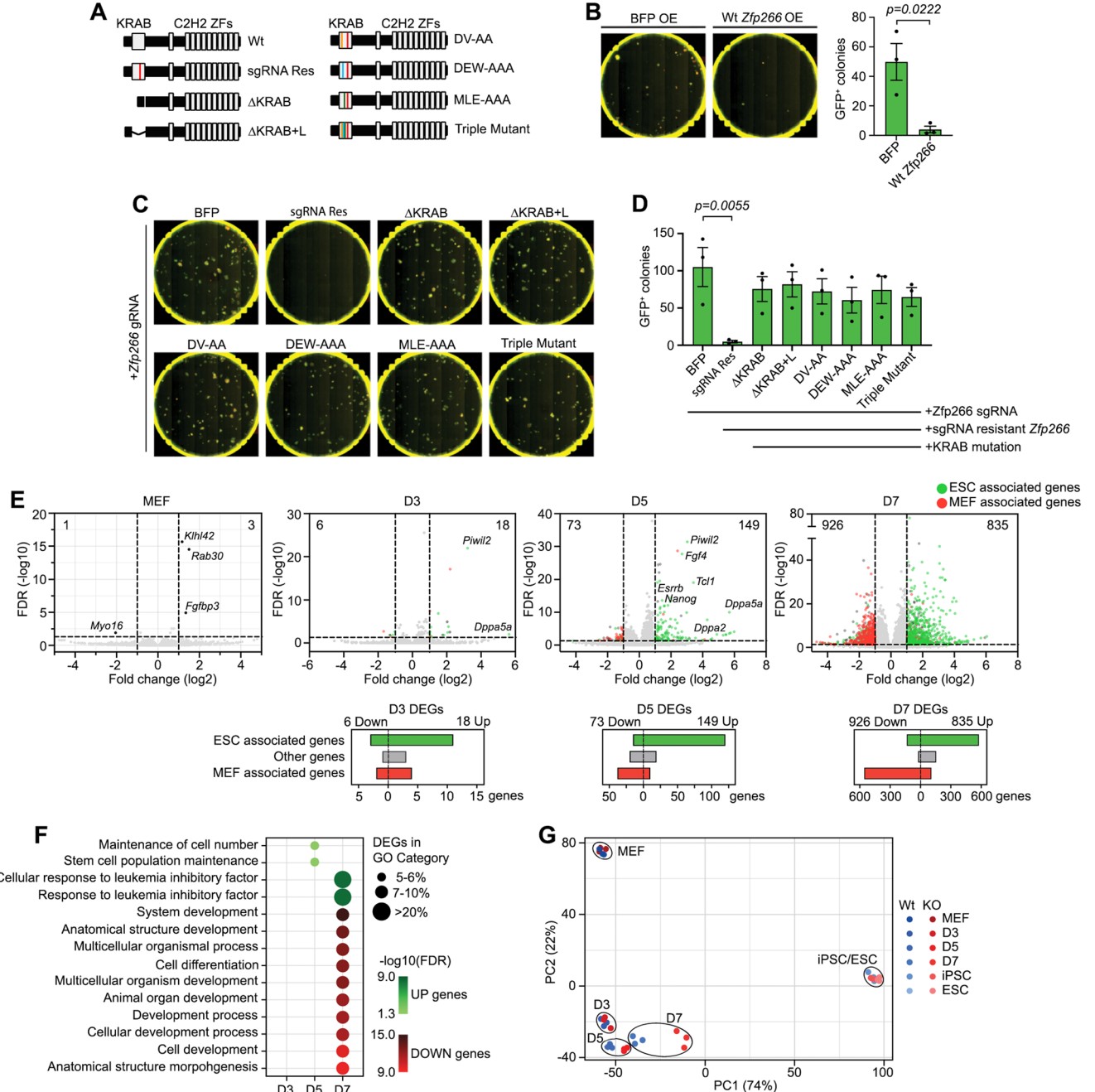

**Fig. 3 | ZFP266 impedes activation of pluripotency genes via its KRAB domain.**
**A** Diagram of *Zfp266* Wt and mutants. A red bar indicates a silent mutation that confers sgRNA resistance. KRAB domain deletion mutants with (ΔKRAB + L) and without a linker (ΔKRAB) do not have the sgRNA target sequence. DV-AA, DEW-AAA, MLE-AAA mutants have alanine substitutions in the indicated critical amino acids in the KRAB domain. Triple mutant contains all the alanine substitutions. **B** *Nanog*-GFP MEF reprogramming with MKOS *piggyBac* transposons and BFP or Wt *Zfp266* cDNA overexpression, imaged at day 15. Red; mOrange, Green; *Nanog*-GFP. Data represent an average of 3 independent experiments. Error bars indicate SEM, significance calculated using an unpaired two-tailed *t*-test. **C** *Cas9 Nanog*-GFP MEF reprogramming with MKOS *piggyBac* transposons, *Zfp266* sgRNA expression as well as cDNA overexpression of BFP, sgRNA resistant *Zfp266*, or sgRNA resistant *Zfp266* with mutations in KRAB domain, imaged at day 15. Red; mOrange, Green; *Nanog*-GFP.

**D** Mean *Nanog*-GFP⁺ colony numbers of **C**. Data represent an average of 3 independent experiments. Error bars indicate SEM, *p*-values calculated using a one-way ANOVA test. **E** RNA-Seq volcano plot of *Zfp266* KO vs Wt MEF, day 3, 5, and 7 of reprogramming. Up-regulated and down-regulated genes in KO cells are shown to the right and left of the plot, respectively (cut-off FDR < 0.05, log2FC >|1|). ESC- and MEF-associated genes (FDR < 0.05, log2FC >|1| in ESCs vs MEFs) are highlighted in green and red. Graphs below volcano plots show the number of ESC-associated, MEF-associated and other genes within D3, D5, D7 reprogramming differentially expressed genes (DEGs). **F** GO enrichment analysis of upregulated and downregulated genes in *Zfp266* KO reprogramming. **G** Principal component analysis of *Zfp266* Wt and KO RNA-Seq samples. Blue dots indicate Wt samples, red dots indicate KO samples, three samples per timepoint. Source data are provided as a Source Data file.

formation of heterochromatin and transcriptional repression[34], suggesting that ZFP266 acts as a suppressor. Consistent with the reprogramming enhancement by *Zfp266* KO, overexpression of exogenous *Zfp266* completely disrupted reprogramming (Fig. 3B). Exogenous overexpression of *Zfp266* mutants either lacking the entire KRAB domain or containing point mutations which disrupt the interaction with KAP-1/TRIM28[35,36] could not abolish *Zfp266* sgRNA-mediated reprogramming enhancement even though the *Zfp266* mutants were resistant to the sgRNA (Fig. 3C, D). This clearly demonstrates that Zfp266 inhibits reprogramming through its KRAB-domain.

To assess further the function of ZFP266, we examined the gene expression changes associated with its depletion (Supplementary Data 3). *Zfp266* gRNA expression in Cas9 TNG MKOS MEFs caused >80% out-of-frame insertion-deletion mutations (indels) (Supplementary Fig. 4A). Nevertheless, RNA-seq of MEFs 4 days after *Zfp266* sgRNA transduction revealed only 4 differentially expressed genes (DEGs) (FDR < 0.05, log2FC > |1|), demonstrating that loss of ZFP266 alone was not sufficient to cause drastic gene expression changes in MEFs (Fig. 3E, Supplementary Data 4). In contrast, the number of DEGs between *Zfp266* KO and wild-type cells rapidly increased during reprogramming from 24 at day 3, to 222 at day 5, and to 1761 at day 7 (Fig. 3E, Supplementary Data 5–7). The majority of DEGs at day 3 and day 5 were upregulated (75% and 67% respectively, Fig. 3E), consistent with the predicted role for *Zfp266* as a transcriptional suppressor. Enhanced up-regulation of pluripotency-associated genes *Piwil2* and *Dppa5a* at day 3, and *Nanog, Esrrb, Dppa2, Tcl1*, etc. at day 5, was already detected in *Zfp266* KO (Fig. 3E). Furthermore, over 60% (11/18), 80% (120/149), 69% (575/835) of up-regulated DEGs in *Zfp266* KO cells at day 3, 5, 7 of reprogramming were genes more highly expressed in ESCs compared to MEFs (FDR < 0.05, log2FC > |1|) (Fig. 3E, green, Supplementary Data 8). Gene ontology (GO) enrichment analysis identified 'stem cell population maintenance' in day 5 and 'response to leukemia inhibitory factor' in day 7 up-regulated DEGs as the most enriched terms (Fig. 3F), while downregulated DEGs at day 7 were significantly enriched in developmental and differentiation terms (Fig. 3F). Principal component analysis (PCA) also indicated that gene expression changes that have occurred in *Zfp266* KO reprogramming at day 5 and 7 reflected an accelerated transition towards a pluripotent state, while *Zfp266* KO iPSCs and ESCs clustered together with wild-type iPSCs/ESCs (Fig. 3G). iPSC clones generated in *Zfp266* KO reprogramming (*Zfp266* KO iPSCs) had comparable pluripotency gene expression, proliferation rate, and an in vitro differentiation capacity to wild-type iPSC clones (Supplementary Fig. 4B–D). Nevertheless, when global gene expression was closely examined, *Zfp266* KO and wild-type iPSCs exhibited 956 DEGs (FDR < 0.05, log2FC > |1|) (Supplementary Fig. 4E). It was in contrast to a relatively small number of DEGs (206, FDR < 0.05, log2FC > |1|) between *Zfp266* KO and wild type ESCs (Supplementary Fig. 4F). Enriched GO terms in the Zfp266 KO/wild type iPSC DEGs were diverse, with some overlap with those in the Zfp266 KO/wild type ESC DEGs (Supplementary Fig. 4G, H). Taken together, these data indicate that *Zfp266* KO enhances and accelerates reprogramming by permitting a more efficient activation of pluripotency genes by OSKM, while this is accompanied by altered expression of some other genes that would not be affected when *Zfp266* is knocked out in a pluripotent state.

## *Zfp266* KO in MEFs results in chromatin opening at B1 SINE-containing ZFP266 binding sites

One possible mechanism by which ZFP266 impedes reprogramming is that it binds and suppresses the pluripotency loci in MEFs and other differentiated cells. To investigate this possibility, we mapped ZFP266 binding sites in MEFs using DamID-seq, which does not required specific antibodies[37,38]. This identified 15,119 unique ZFP266 binding sites (Fig. 4A), predominantly situated in introns or intergenic regions (Supplementary Fig. 5A, Supplementary Data 9). These ZFP266 binding sites have low chromatin accessibility as measured by ATAC-seq in MEFs, 72 h after reprogramming, as well as in iPSCs[39] (Fig. 4A). ZFP266 binding sites were predominantly enriched for somatic AP-1 TF motifs (Fig. 4B), and little OSKM binding was observed at the same loci in 48 hr reprogramming or ESC ChIP-Seq datasets[40,41] (Supplementary Figure 5B, C). This disputed the idea that ZFP266 functions as a suppressor at the pluripotency-related gene loci in differentiated cells. Thus, we investigated whether any TE families were enriched in ZFP266 binding sites, since many KRAB-ZFPs are known to bind and suppress transcription of TEs[15]. In line with this, we found the 15,119

ZFP266 DamID-seq peaks in MEFs were highly enriched in SINEs with about two-thirds (10,523) overlapping with SINEs (Fig. 4C, D). Of SINE subfamilies, B1 SINEs in particular exhibited both the most significant enrichment and the most abundant overlap with ZFP266 binding sites (Fig. 4C, D). Furthermore, de novo motif analysis of ZFP266 binding sites identified 3 long de novo motifs which all corresponded to parts of the B1 SINE consensus sequence (Fig. 4E), suggesting that ZFP266 might bind B1 SINEs.

We next examined how depletion of ZFP266 might affect chromatin accessibility. To this end, we performed ATAC-Seq of *Zfp266* KO MEFs and identified 479 more open regions (MORs) compared to WT MEFs, while only one locus was found to be a more closed (Fig. 4F and Supplementary Data 10). Considering the predicted suppressor function of ZFP266, we next examined whether ZFP266 binds to the MORs in wild-type MEFs. Although only about 25% (123/479) of *Zfp266* KO MEF MORs overlapped with ZFP266 DamID-seq peaks, non-overlapped MORs also had increased DamID-seq signals albeit at a lower level (Fig. 4G), unlike randomly selected control regions with similar chromatin accessibility (Supplementary Fig. 5D). This suggests that more than 25% of MORs are likely bound by ZFP266, while they were not identified as a 'peak' with our DamID-seq peak calling pipeline due to the cut-off criteria and/or technical limitations. In fact, similar to ZFP266 DamID-seq peaks in MEFs, *Zfp266* KO MEF MORs were mainly located in intergenic regions and introns (Supplementary Fig. 5E), and enriched in AP-1 TF motifs, with 87% of all *Zfp266* KO MEF MORs containing at least one AP-1 TF motif (Fig. 4H). De novo motif discovery analysis also identified 5 motifs that overlap with the B1 SINE consensus sequence (Fig. 4I), consistent with the fact that SINEs were the most enriched repetitive element (Fig. 4J), and 92% (441/479) of *Zfp266* KO MEF MORs had at least one SINE (Fig. 4K). Overall, these data suggest that ZFP266 binds to B1 SINEs in MEFs to keep target loci closed, and removal of ZFP266 allows TFs that binds nearby, like AP-1 TFs, to facilitate chromatin opening (Fig. 4L). However, ZFP266 does neither bind to nor regulate pluripotency gene loci in MEFs.

## *Zfp266* KO in reprogramming results in chromatin opening at B1 SINE-containing OSK binding sites

In order to address why *Zfp266* KO results in significant reprogramming enhancement, we performed ATAC-seq 72 h after reprogramming with and without *Zfp266* KO. Similar to the KO effects in MEF, *Zfp266* KO reprogramming cells exhibited 1522 MORs, the majority of which were situated in intergenic regions and introns (Supplementary Fig. 6A), with only 86 more closed regions compared to wild-type cells (Fig. 5A, Supplementary Data 11). They were also significantly enriched in SINEs, particularly B1 SINEs (Fig. 5B), with >90% (1459/1522) of MORs containing at least one SINE (Fig. 5C). De novo motif discovery analysis also identified motifs that correspond to the B1 SINE consensus sequence as the most significant motifs (Fig. 5D). However, these loci hardly overlapped with *Zfp266* KO MEF MORs (Fig. 5E), suggesting a context dependency for which loci become more open in the absence of ZFP266. The overlap with MEF ZFP266 DamID-seq peaks was also minimal, with only ~10% of MORs overlapping (Supplementary Fig. 6B), and non-overlapped MORs did not have increased DamID-seq signals compared to the control regions with a similar chromatin accessibility (Supplementary Fig. 6B). This indicated that upon OSKM expression, ZFP266 changes binding sites at which it regulates chromatin accessibility. TF motif enrichment analysis revealed that KLF, SOX, and the OCT4::SOX2 motifs were highly enriched in *Zfp266* KO reprogramming MORs, particularly with KLF family (KLF1, KLF5, KLF4, KLF9, KLF12) motifs identified in >90% of these MORs, while AP-1 TF motifs were also enriched (Fig. 5F). Next, we classified *Zfp266* KO reprogramming MORs into two groups using *K*-means clustering (Fig. 5G). The first cluster (121 regions) are open in both wild-type and *Zfp266* KO MEFs, and then becomes more closed upon reprogramming, while *Zfp266* KO cells are more resistant to this closing (cluster 1, Fig. 5G). The second cluster

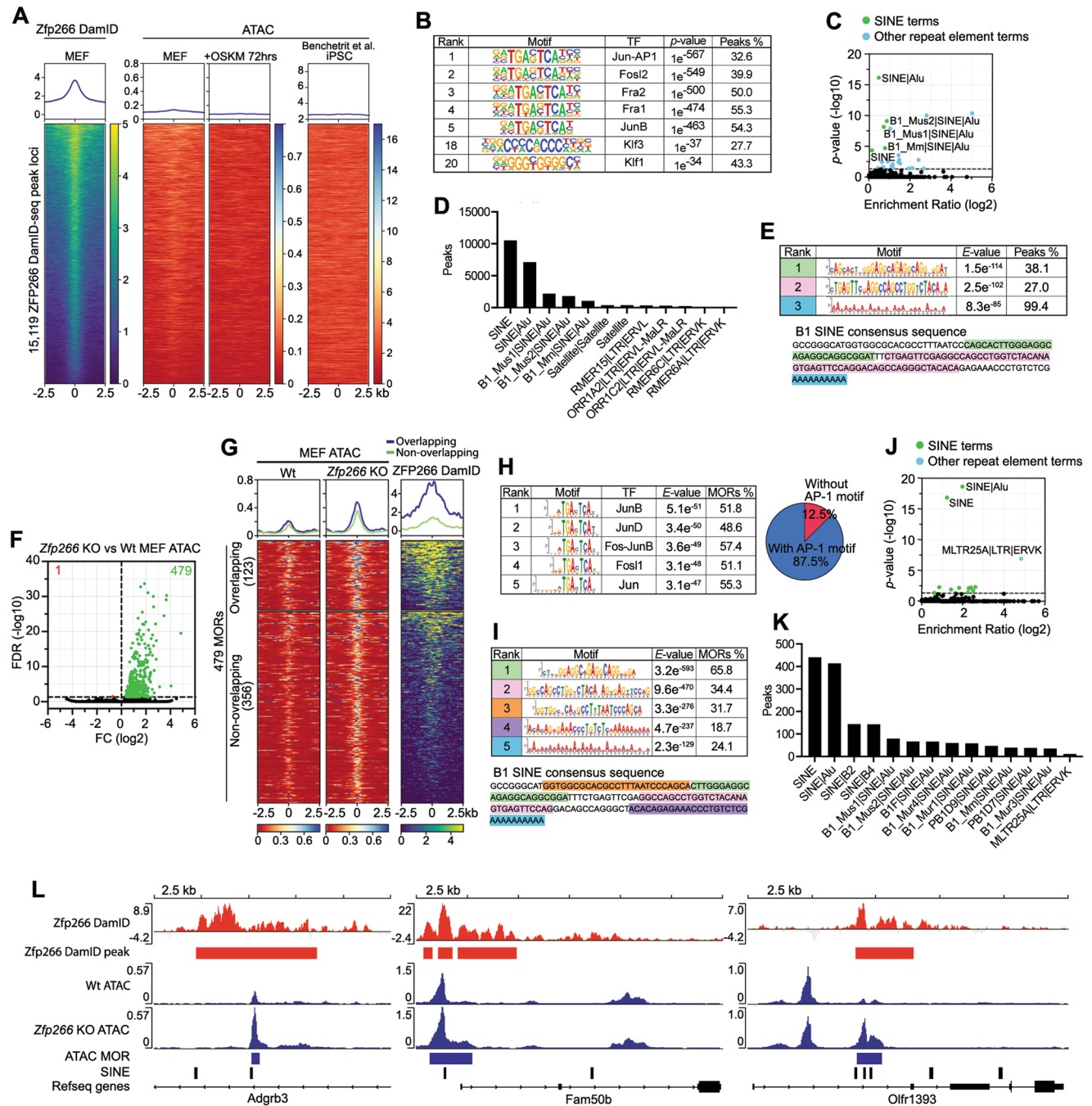

**Fig. 4 | *Zfp266* KO in MEFs results in chromatin opening at the SINE-containing ZFP266 binding sites. A** ZFP266 DamID-seq signals in MEFs, ATAC-seq signals in MEFs, +72 h of reprogramming, and iPSCs at the ZFP266 DamID-seq peak loci. **B** Motif enrichment analysis with HOMER on ZFP266 DamID-seq peaks. **C** Significance and fold enrichment ratio of transposable element (TE) families overlap with Dam-ZFP266 peaks. Green dots indicate significantly enriched SINEs, blue dots indicate other significantly enriched TEs. The *p*-values were calculated by the cumulative hypergeometric distribution (a one-tailed test) in **B** and **C. D** Number of ZFP266 DamID-seq peaks that overlap with TEs. **E** De novo motif discovery analysis with MEME on ZFP266 DamID-seq peaks. The identified motifs correspond to parts of the B1 SINE consensus sequence, indicated by matching colors. **F** Volcano plot of *Zfp266* KO vs Wt MEF ATAC-seq. Green and red dots indicate more open regions (MORs) and more closed regions in *Zfp266*

KO MEFs, respectively (FDR < 0.05). **G** ATAC-seq and ZFP266 DamID-seq signals in the *Zfp266* KO MEF MORs, overlapped (top) and non-overlapped (bottom) with ZFP266 DamID peaks in MEFs. **H** Motif enrichment analysis on *Zfp266* KO MEF MORs. Percentages of MORs containing each motif and AP-1 motif are indicated. **I** De novo motif discovery analysis on *Zfp266* KO MEF MORs. The top five most significant motifs correspond to parts of the B1 SINE consensus sequence, indicated by matching colors. **J** Significance and fold enrichment ratio of transposable element (TE) families overlap with *Zfp266* KO MEF MORs. Green dots indicate significantly enriched SINEs, blue dots indicate other significantly enriched TEs. The *p*-values were calculated by the cumulative hypergeometric distribution. **K** Number of *Zfp266* KO MEF MORs that overlap with TEs. **L** Examples of *Zfp266* KO MEF MORs (Blue) with SINE (black), overlapping with ZFP266 DamID-seq peaks (red).

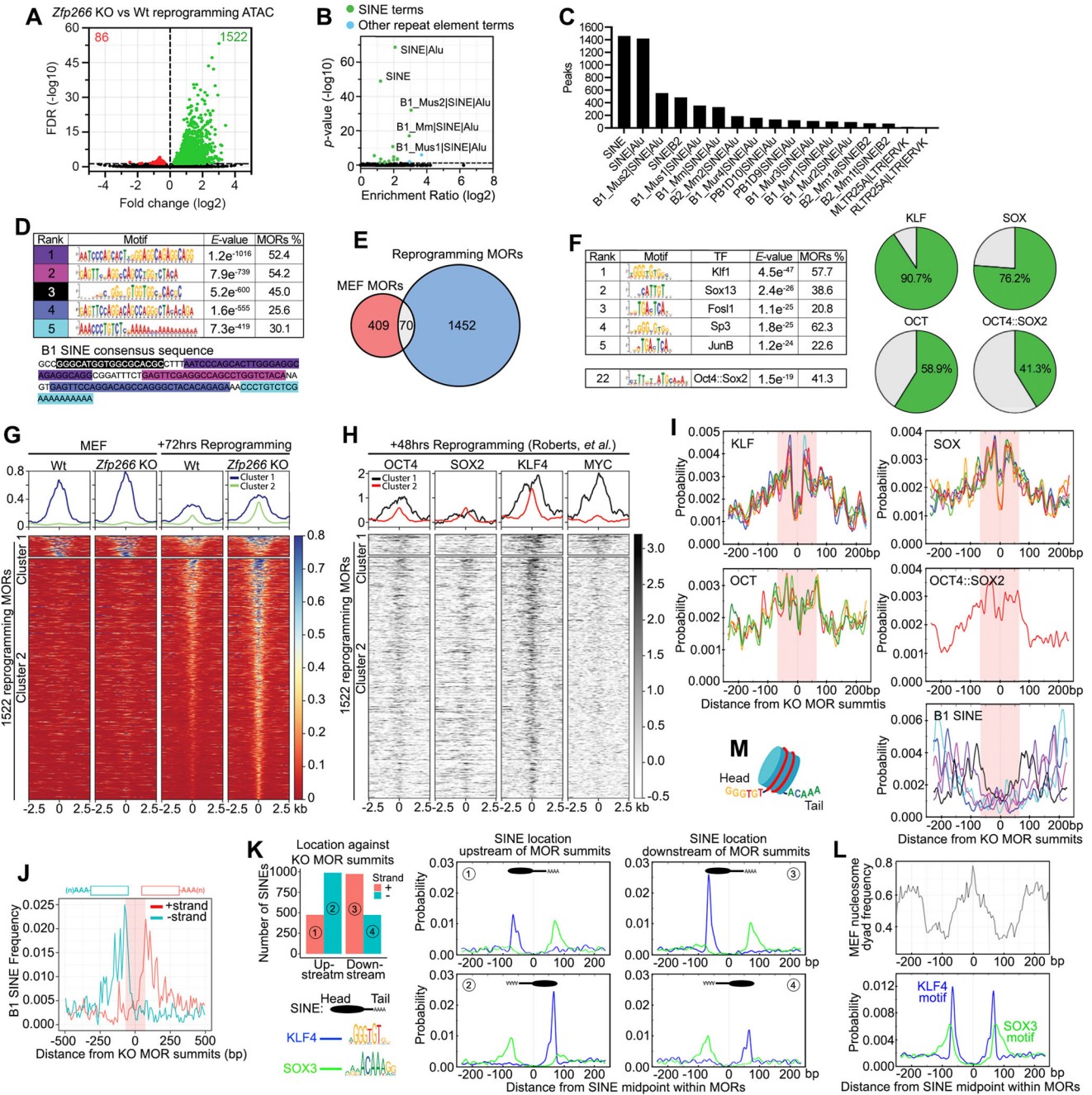

**Fig. 5 | Reprogramming *Zfp266* KO MEFs results in chromatin opening at OSK bound, B1 SINE containing loci. A** ATAC-seq volcano plot of *Zfp266* KO vs Wt reprogramming at 72 h. Green and red dots indicate more open regions (MORs) and more closed regions in *Zfp266* KO reprogramming, respectively (FDR < 0.05). **B** Significance and fold enrichment ratio of transposable element (TE) families within Zfp266 KO MEF MORs. Green dots indicate significantly enriched SINEs, blue dots indicate other significantly enriched TEs. The p-values were calculated by the cumulative hypergeometric distribution. **C** Number of *Zfp266* KO reprogramming MORs with TEs. **D** De novo motif discovery analysis on *Zfp266* KO reprogramming MORs. The motifs correspond to parts of the B1 SINE consensus sequence are indicated by matching colors. **E** Overlap between *Zfp266* KO MEF and *Zfp266* KO reprogramming MORs. **F** Motif enrichment analysis with *Zfp266* KO reprogramming MOR summits and percentages of MORs with KLF, SOX, OCT (POU) family and OCT4:SOX2 motifs. **G** Classification of *Zfp266* KO reprogramming MORs based on ATAC-seq signals in MEF and reprogramming 72 h. **H** Reprogramming 48 h OSKM ChIP-Seq heatmap at *Zfp266* KO reprogramming MORs. **I** KLF, SOX, OCT (POU) family, OCT4:SOX2 and SINE motif distribution within *Zfp266* KO reprogramming MORs. 70 bp from the summit is highlighted in pink. The colors of B1 SINE motifs correlate to those in **D**. **J** Orientation-biased distribution of B1 SINEs within *Zfp266* KO reprogramming MORs. Head of B1 SINE tends to locate on the MOR summit side. 70 bp from the summit is highlighted in pink. **K** Numbers of SINEs located on the plus (1, 3) or minus (2, 4) strand either upstream (1, 2) or downstream (3, 4) of the MOR summits in reprogramming MORs (left top). KLF4/SOX3 motif enrichment in each group with midpoint of B1 SINE in the centre (right four panels). **L** Nucleosome dyad frequency at B1 SINEs within reprogramming MORs using MNase-seq data with MEFs (GSM1004654) (top) and KLF4 and SOX3 motif enrichment at the same regions (bottom). **M** A model of nucleosome wrapped by B1 SINE with KLF and SOX motifs at the head and tail.

contained the majority (1401) of *Zfp266* KO reprogramming MORs, which are closed in both wild-type and *Zfp266* KO MEFs, and become open following reprogramming, an effect which is enhanced when *Zfp266* is knocked out (cluster 2, Fig. 5G). Cluster 2 indicates that removal of ZFP266 facilitates reprogramming factor-mediated chromatin opening. In fact, we observed OSK binding in cluster 2 reprogramming MORs with particularly strong KLF4 signal using published reprogramming 48-h ChIP-Seq datasets[41] (Fig. 5H). Similar OSK binding was observed in an ESC ChIP-Seq dataset albeit to a lesser extent[40] (Supplementary Fig. 6C), and about one third of the cluster 2 loci have an open chromatin state in iPSCs (Supplementary Fig. 6D), suggesting that some of the MORs are OSK targets in pluripotent cells. Interestingly, while both OSK binding and motifs were enriched close to the MOR peak summit (within 70 bp) (Fig. 5H, I), SINEs were depleted from summits and were instead enriched immediately upstream or downstream (~70 bp away from the summit) (Fig. 5I, J). Furthermore, B1 SINEs within the MORs had an orientation bias such that the 5′ head sequence was positioned inwards facing towards the peak summit (Fig. 5I, J). This positional and directional bias within the MOR was exclusive to B1 SINE subfamilies as B2 SINEs exhibited no such bias (Supplementary Fig. 6E). In order to make a careful analysis, we separated B1 SINEs within *Zfp266* KO reprogramming MORs into 4 groups based on its orientation and location; 1) on the positive strand and up-stream of MOR summits, 2) on the negative strand and up-stream of MOR summits, 3) on the positive strand and down-stream of MOR summits, and 4) on the negative strand and down-stream of MOR summits (Fig. 5K). When motif enrichment analysis was carried out with the midpoint of these ~140 bp B1 SINEs in the centre, it became clear that KLF and SOX motifs were enriched at/near the head and tail of B1 SINE, respectively (Fig. 5K). As the head and tail of B1 SINE are rich in G, and the tails are rich in A, it is possible that mutations in the B1 SINE sequence have resulted in the generation of KLF and SOX motifs, which are rich in G and A, respectively. The size of B1 SINE (~140 bp) is similar to the length of DNA wrapping the nucleosome core (~147 bp). When we examined nucleosome position against the B1 SINEs in *Zfp266* KO reprogramming MORs, we found that the midpoint of B1 SINEs were highly enriched in the nucleosome dyad in MEFs, indicating that nucleosomes are well-positioned in B1 SINEs. Interestingly, this means KLF/SOX motifs are enriched at the nucleosome DNA entry-exit sites (Fig. 5L, M). This feature was not specific to B1 SINEs within MORs and could be observed using all B1 SINEs, while enrichment of KLF4 motif was less prominent (Supplementary Fig. 6F). Notably, primate-specific Alu elements, which are closely related to rodent B1 SINEs, are also known to have strong association with nucleosomes[42]. Based on these data together with a report that somatic TFs' binding sites drastically change upon OSKM expression[40], we speculate that in response to reprogramming factor binding at KLF/SOX binding motifs in the head/tail of B1 SINEs, ZFP266 is recruited to those B1 SINEs during reprogramming, where it then acts to impede chromatin opening.

## Facilitating chromatin opening at ZFP266 targeted SINEs enhances reprogramming

In order to validate the binding of ZFP266 to SINEs, we generated an activator version of ZFP266 with the KRAB domain replaced by a flexible linker and three transactivating domains VP64, p65, and Rta (VPR)[43] (Fig. 6A), and performed luciferase reporter assays using HEK293T cells. Enhanced luciferase expression was observed when VPR-*Zfp266*, but not blue fluorescent protein (BFP) or VPR only controls, was co-transfected with a reporter plasmid containing the B1 SINE consensus sequence upstream of a SV40 minimal promoter (Fig. 6B). Co-expression of wild-type *Zfp266* alongside VPR-*Zfp266* attenuated this reporter expression (Fig. 6C), confirming ZFP266 specifically binds B1 SINEs. We next examined whether VPR-ZFP266 can bind to *Zfp266* KO reprogramming MORs using the Luciferase assay.

We selected SINE containing MORs in three genes, *B3gnt3*, *Piwil2*, and *Snx20*, whose transient up-regulation during reprogramming was significantly augmented by *Zfp266* KO (Fig. 6D). These loci are closed in MEFs, open up more in *Zfp266* KO cells upon reprogramming, and are bound by KLF4 at 48 h of reprogramming (Fig. 6E). Each MOR was cloned upstream of a minimal SV40 promoter in both a forward and reverse orientation in a luciferase reporter vector. We found that VPR-*Zfp266* could also enhance luciferase expression from these MOR reporter vectors (Fig. 6F), while co-expression of wild-type *Zfp266* ablated it (Fig. 6G). Deleting B1 SINE sequences from the *B3gnt3*, *Piwil2* and *Snx20* MORs diminished VPR-*Zfp266*'s ability to enhance luciferase expression, confirming that ZFP266 binds to reprogramming MORs specifically via B1 SINE sequences (Fig. 6H–J). We also confirmed that OSKM expression enhanced luciferase expression from the *B3gnt3* and *Snx20* MOR containing reporter vectors in MEFs (Fig. 6K), indicating that the MORs have OSKM-dependent regulatory activity. In addition, removing B1 SINE sequences from the *Snx20* MOR led to an increase in Luciferase expression following OSKM induction (Fig. 6L). While luciferase assays do not always reflect endogenous gene regulation, these data indicate that B1 SINEs function to repress reprogramming factor-mediated transactivation via ZFP266 binding. Finally, overexpression of the VPR-*Zfp266* together with OSKM led to accelerated and enhanced reprogramming with a robust appearance of *Nanog*-GFP+ colonies by day 9 (Fig. 7A, B). Taken together, we propose a model where 1) ZFP266 binds to B1 SINEs adjacent to KLF4/SOX2 binding motifs during reprogramming and acts to impede chromatin opening, and 2) KO of *Zfp266* (or recruitment of co-activators to these loci) tips the balance in favor of OSK, allowing them to establish a more open chromatin state to drive gene activation necessary for successful reprogramming (Fig. 7C).

In order to evaluate the possibility that this model acts on pluripotency gene induction, we looked into the positions of SINEs related to iPSC open chromatin loci. Of the 178,112 iPSC ATAC-seq peaks, 22% (39,353) had B1 SINEs, which were mostly closed in MEFs (Supplementary Fig. 7A). ~70 bp from the midpoint (i.e. head and tail regions) of these B1 SINEs are enriched in KLF and SOX motifs (Supplementary Fig. 7B), resulting in 21,150 iPSC ATAC-seq peaks with at least one B1 SINE and one KLF/SOX motif. 76 out of 137 genes with a GO term "stem cell population maintenance" were associated with such ATAC-seq peaks, including *Nanog, Esrrb*, and *Dppa2* (Supplementary Fig. 7C, D). Up-regulation of these pluripotency genes during reprogramming could be facilitated by lack of B1 SINE/ZFP266-mediated chromatin closing in *Zfp266* KO cells.

In order to investigate whether roles ZFP266 is conserved in human cell reprogramming, we first sought to identify the human orthologue of murine ZFP266. ZNF266, ascribed as the potential orthologue of ZFP266, displays a relatively low protein sequence similarity to ZFP266. However, two other KRAB-ZFPs, ZNF561, and ZNF562, have higher similarities (Supplementary Figures 8A, B), with ZNF561 having ~70% homology at the residues that specify target sequence (amino acids at positions −1, 2, 3, 6 within the ZF). Among those, only the overexpression of *ZNF561* led to the inhibition of mouse iPSC generation in the absence of endogenous *Zfp266* in a KRAB domain-dependent manner Supplementary Fig. 8C–F), indicating that ZNF561 is a possible orthologue of mouse ZFP266. Next, we examined if ZFP266 can bind to Alu elements in the human genome, which are closely related to Mouse B1 SINEs but have 2 7SL RNA-like sequences (Supplementary Fig. 8G). When the B1 SINE within the *Snx20* reporter was replaced by Alu, *VPR-Zfp266* could induce the reporter expression (Supplementary Fig. 8H), suggesting that ZFP266, and possibly ZNF561, can bind to Alu sequences. Nevertheless, *VPR-Zfp266* did not enhance human iPSC generation by OSKM (Supplementary Fig. 8I). Interestingly, only 5.9% of ATAC-seq peaks in human ESCs (9,210/155,757) contain an Alu sequence (Supplementary Fig. 8J), although KLF and SOX family motif

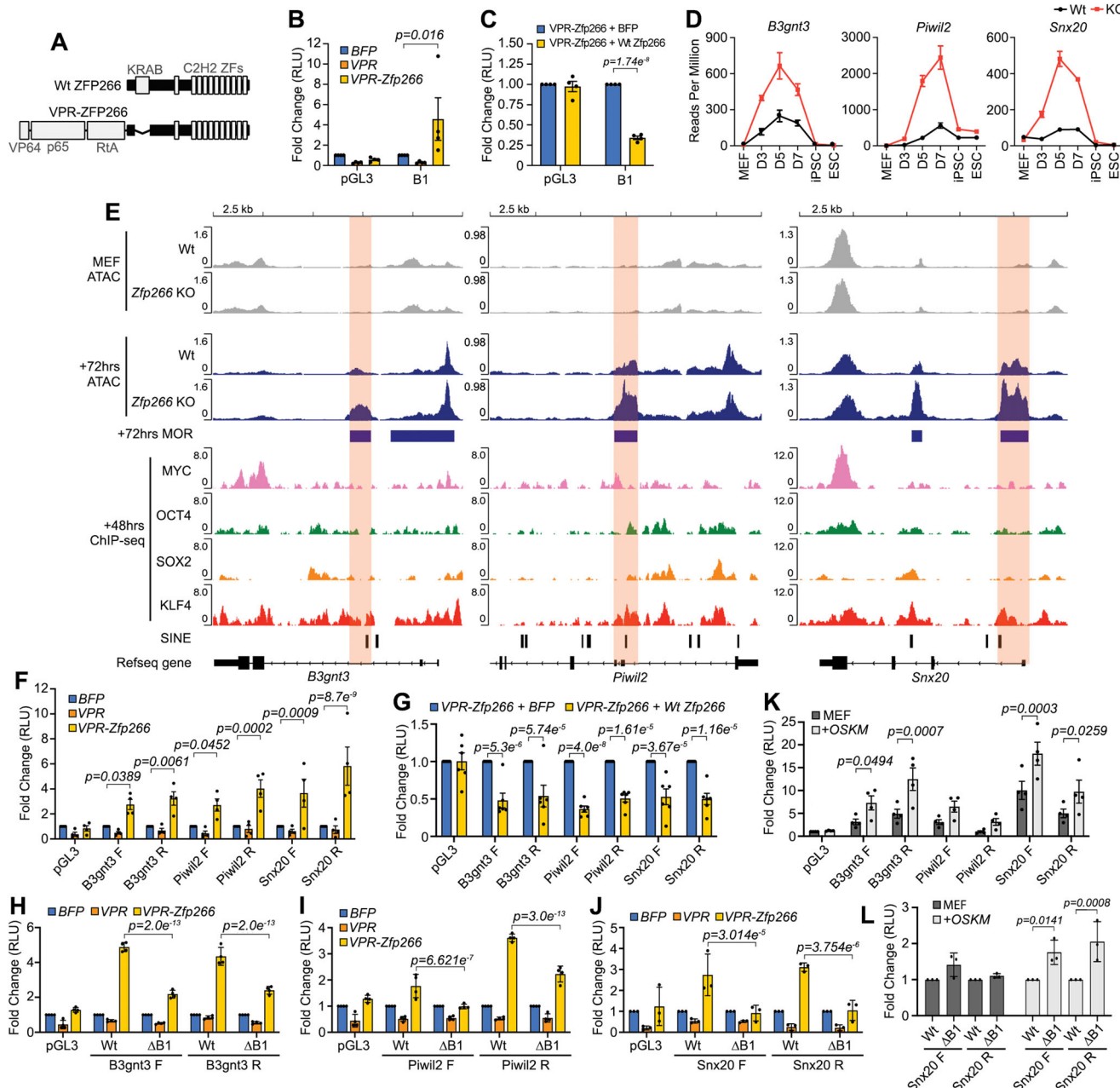

**Fig. 6 | ZFP266 binds to B1 SINEs in Zfp266 KO reprogramming MORs to impede OSKM-mediated gene activation. A** Diagram of Wt ZFP266 and a synthetic activator version of ZFP266, VPR-ZFP266. **B, C** Luciferase reporter assay with either an empty reporter vector pGL3 or with a reporter vector with the B1 SINE consensus sequence, co-expressed with either BFP, VPR only or VPR-*Zfp266* expression vectors (**B**), with either BFP or Wt *Zfp266* expression vectors in the presence of VPR-ZFP266 (**C**) in HEK293T cells. Data in B and C represent an average of 4 independent experiments. Error bars indicate SEM. RLU: Relative Light Units, significance calculated using a two-way ANOVA test. **D** *B3gnt3*, *Piwil2*, and *Snx20* mRNA expression from the *Zfp266* Wt and KO reprogramming RNA-seq data with 3 independent samples. **E** ATAC-seq, ChIP-seq signals at the *B3gnt3*, *Piwil2*, and *Snx20* MORs, cloned in both forward (F) and reverse (R) directions (relative to gene orientation) for luciferase reporter assays (highlighted in orange). Input subtracted ChIP-seq data are shown. **F–J** Luciferase reporter assay with an empty reporter vector pGL3 or vectors containing *B3gnt3*, *Piwil2*, and *Snx20* MORs co-transfected with either BFP, VPR only or VPR-*Zfp266* expression vectors (**F**), co-transfected with either BFP or Wt *Zfp266* expression vectors in the presence of VPR-ZFP266 (**G**), an empty reporter vector pGL3, vectors containing *B3gnt3* (**H**), *Piwil2* (**I**) and *Snx20* (**J**) MORs with (ΔB1) or without (Wt) B1 SINE deletion co-transfected with either BFP, VPR only or VPR-*Zfp266* expression vectors in HEK293T cells. **K, L** Luciferase reporter assay with empty reporter vector pGL3 or vectors containing *B3gnt3*, *Piwil2* and *Snx20* MORs (**K**), *Snx20* MOR reporter with (ΔB1) or without (Wt) B1 SINE deletion (**L**), using MEF with or without OSKM expression. **F–L**. Data represent an average of 3 (**J** and **L**), 4 (**F, H, I,** and **K**), and 6 (**G**) independent experiments. Error bars indicate SEM. *p*-values calculated using a two-way ANOVA test. Source data are provided as a Source Data file.

enrichment were observed near the beginning of the 2nd 7SL-like sequence and at the end of these Alu within hESC ATAC-seq peaks, respectively (Supplementary Fig. 8K). In contrast, >20% of ATAC-seq peaks in mouse iPSCs (39,353/178,112) have B1 SINE, many of which are associated with pluripotency genes (Supplementary Fig. 7). We speculate that the non-conserved position of B1 SINE and

Alu elements in the genome is in part contributing to *VPR-Zfp266* inability to enhance human iPSC generation.

## Discussion

Reprogramming towards iPSCs is a conflict between OSKM transcription factors trying to establish a pluripotent state and somatic factors

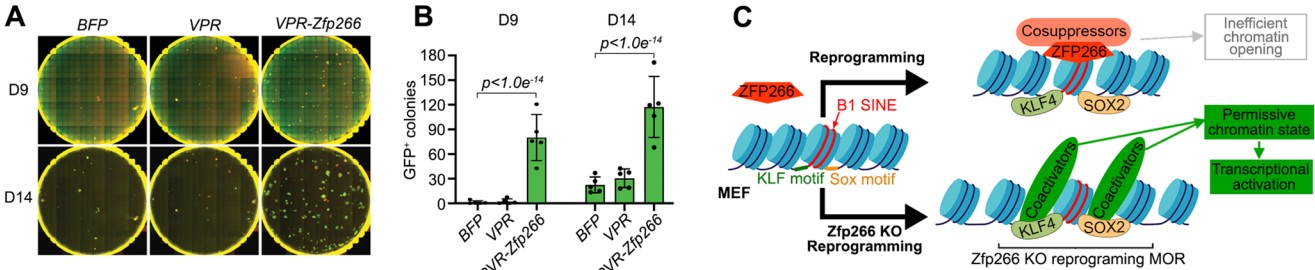

**Fig. 7 | Chromatin accessibility at the ZFP266 target loci affects reprogramming efficiency. A** Day 9 and day 14 after OSKM induction with overexpression of either BFP, *VPR* only or *VPR-Zfp266*. Red; mOrange, Green; *Nanog*-GFP.
**B** Quantification of *Nanog*-GFP⁺ colony numbers at day 9 and day 14. Data represents an average of three independent experiments. Error bars indicate SEM. *p*-values calculated using a one-way ANOVA test. **C** Mechanistic model depicting how Zfp266 KO enhances reprogramming. Source data are provided as a Source

Data file. B1 SINEs are strongly associated with nucleosomes and often have KLF and SOX motif at the head and tail due to **G** and **A** rich sequence, respectively. ZFP266 binds to some of SINEs in a cell type- and context-dependent manner. Upon reprogramming factor expression, ZFP266 is recruited to B1 SINE with KLF/SOX motifs at its head or tail, respectively, and impedes chromatin opening (top). Zfp266 KO results in increased chromatin accessibility in those loci, facilitating pluripotency gene expression.

trying to resist this disruption in cell identity. Our CRISPR screen identified 24 reprogramming roadblock genes, whose depletion facilitates OSKM-mediated pluripotency induction, including 8 previously reported roadblock genes. One of the most robust roadblock genes, *Zfp266*, is recruited to OSK binding sites through the recognition of adjacent B1 SINEs, where it impedes chromatin opening via its KRAB domain. This probably underlays the accelerated up-regulation of pluripotency gene by OSKM in both *Zfp266* KO and VPR-*Zfp266* overexpression reprogramming. In fact, many of *Zfp266* KO MORs at 72 h of reprogramming have an open chromatin state and bound by OSK in iPSCs/ESCs, and several of them are associated with pluripotency genes or other genes highly expressed in ESCs, including *Pou5f1, Sall4, Zfp42, Klf2, Piwil2, Fbxo15, Dnmt3l,* and *Tet1/2* (Supplementary Data 12). These genes were more efficiently up-regulated in *Zfp266* KO reprogramming compared to the control (Supplementary Data 3). In addition, more than 20% of iPSC open chromatin loci contained B1 SINEs, of which more than half had at least one KLF/SOX motif (Supplementary Figure 7). Therefore, depleting Zfp266 likely helps to accelerate the establishment of a pluripotent chromatin state. Loss of ZFP266 in MEFs resulted in only 4 DEGs, suggesting that ZFP266 may not play a significant role in a static state but act as a safeguard against drastic changes of cellular states mediated by newly expressed TFs and/or extracellular cues, such as cytokines. During early embryo development, i.e. 2–8 cell stage, ICM, and ESCs, SINEs, particularly B1 SINEs, are enriched in the open chromatin regions, but not in subsequent developmental stages[14]. The data from International Mouse Phenotyping Consortium shows that only 6.5% of pups from *Zfp266* heterozygous intercrosses are homozygous for the KO allele, presenting incomplete penetrance and suggesting possible roles of ZFP266 during development along with compensation mechanisms (https://www.mousephenotype.org/data/genes/MGI:1924769). Further investigation might reveal the roles of ZFP266 in B1 SINE region closing and regulation of neighboring genes during embryo development. A recent publication identified enrichment of B1 SINEs at the flanks of CD8 + T-cell specific enhancers[44]. Our analyzes revealed that B1 SINEs are strongly associated with nucleosomes, similar to Alu and LINE retrotransposons in human[42]. Moreover, we identified that B1 SINE-associated nucleosome DNA entry-exit sites are enriched in particular TF recognition motifs, including KLF and SOX family motifs. Thus, the positional bias of SINEs against regulatory regions/enhancers could be a much more general feature of B1 SINE, and SINEs could affect chromatin states via KRAB-ZFPs, while it could also be influenced by surrounding DNA sequences and/or other proteins. KLF4 has been shown to bind primate/human specific TEs in naive human ESCs and during reprogramming[13,19]. While B1 SINEs are restricted to rodents, KLF4 binding is enriched in the old world monkey-, ape-, and human-specific

TEs, HERVH, HERVK, and SVA, in naïve human ESCs[19]. The enrichment of KLF4 binding in *Zfp266* KO reprogramming MORs with SINEs indicates the conserved function of KLF4 to regulate gene expression via TE-containing regions in reprogramming/pluripotency across species, which is of clear interest for further investigation. It has been reported that ~2/3 of human KRAB-ZFPs bind to TEs genome-wide, and KRAB-ZFPs suppress not only TEs, but also an expression of genes nearby[16,17]. Our results suggest a possibility that other KRAB-ZFPs could act as barriers in different TF-mediated cell conversions or differentiation of pluripotent cells to specific cell types, and therefore elimination of those obstacles or the use of engineered activator version of KRAB-ZFPs might realize more efficient cellular identity changes. Our CRISPR/Cas9-mediated genome-wide KO screening also identified several other genes whose inhibition of pluripotency induction would have never been predicted from transcriptomic analyzes. Further understanding how these genes hamper OSKM-mediated reprogramming will bring us a better understanding of how to control cellular identities.

## Methods
### Cell culture
MEFs were cultured in MEF medium (Glasgow minimum essential medium [GMEM] supplemented with 10% fetal calf serum [FCS], 100 U/ml penicillin-streptomycin, 13 non-essential amino acids (Invitrogen), 1 mM sodium pyruvate, 2 mM glutamine, 0.05 mM 2-mercaptoethanol (Life Technologies) supplemented with 5 ng/ml fibroblast growth factor-2 (FGF2) and 1 ng/ml heparin). ESCs were cultured in ESC medium (MEF medium without FGF2 and heparin, supplemented with human LIF [leukemia inhibitory factor], 100 U/ml), as described previously[20]. Reprogramming was performed in reprogramming medium (ESC medium supplemented with 300 ng ml⁻¹ of doxycycline (Sigma) and 10 μg ml⁻¹ of L-ascorbic acid or 2-Phospho-L-ascorbic acid trisodium salt (Vitamin C) (Sigma). NSCs were cultured in NSC complete medium consisting of DMEM/F-12 Media, 1:1 Nutrient Mixture (Sigma), 1X N2 supplement (Thermo Fisher Scientific), 1X B27 supplement (Thermo Fisher Scientific), 8 mM glucose (Sigma), 100 U ml⁻¹ Penicillin/Streptomycin (ThermoFisher Scientific), 0.001% Bovine Serum Albumin (BSA) (ThermoScientific), 0.05 mM β-mercaptoethanol (Thermo Fisher Scientific), supplemented with 10 ng ml⁻¹ mouse Epidermal growth factor (EGF) (Peprotech) and 10 ng ml⁻¹ human FGF2 (Peprotech).

### Plasmids
Plasmids used in this work are summarized in Supplementary Data 13. The plasmids and their sequences are available upon request.

## Generation of Cas9 TNG MKOS ESC line and Cas9 TNG MKOS MEFs

The *Rosa26* targeting vector carrying EF1α-hCas9-ires-neo cassette (Addgene #67987) was electroporated into TNG MKOS ESC line derived from E14Tg2a mouse ESC line[20,24]. Correct targeting was confirmed by Southern blotting using KpnI and MscI digested genome DNA for a 5′ and 3′ probe, respectively. The 5′ and 3′ probes were generated from PCR amplicons using the following primers, 5′ forward CAAGTGCTCCATGCTGGAAGGATTG, 5′ reverse TGATTGGGGAGGAT CCAGATGGAG, 3′ forward GGATTGCACGCAGGTTCTCCG 3′ reverse CGCCGCCAAGCTCTTCAGCAA and genome DNA (for 5′ probe) or the targeting vector (for 3′ probe) as a template. Cas9 TNG MKOS MEFs were isolated from E12.5 chimeric embryos generated via morula aggregation and the proportion of transgenic MEFs from each embryo was assessed by measuring % of mOrange⁺ cells after exposing 1/10th of the dissociated cells to Dox for 2 days as described previously[20]. All animal experiments were approved by the University of Edinburgh Animal Welfare and Ethical Review Body, performed at the University of Edinburgh and carried out according to regulations specified by the Home Office and Project License. Mice were kept under 12 h of light (7 am − 7 pm) and 12 h of dark (7 pm−7 am) cycles.

## sgRNA screen

The sgRNA library (Addgene #67988) was prepared as described before[24] $9 \times 10^6$ high contributions (>98% mOrange⁺ 2 days after addition of dox) TNG MKOS Cas9 MEF plated across 90 wells of 6-well culture plates were exposed to lentiviral sgRNA library at MOI = 2 for 4 h. We used MOI = 2 (infection efficiency ~86%) in order to increase the coverage of the sgRNA library, presuming the scarcity of reprogramming relevant genes and the negligible probability of the same neutral sgRNAs being repeatedly present in combination with relevant sgRNAs. After viral-containing media was removed, the cells were cultured in 3 ml of reprogramming medium. The medium was replaced once 3 ml a day for the first 3 days, and then twice 4 ml a day from day 4 of reprogramming. From day 8, the media was switched to ESC medium supplemented with puromycin (1 μg ml⁻¹) and the medium was replenished twice a day with 4 ml / well until day 16. Puromycin-resistant, *Nanog*-GFP⁺ cells were then sorted using the FACS AriaII (BD Biosciences) and stored at −80 °C as cell pellets before extraction of genomic DNA. The screening was performed in triplicate. Genomic DNA from $3 \times 10^7$ sorted GFP⁺ cells was extracted using the Blood & Cell Culture DNA Maxi Kit (Qiagen). Amplification of sgRNA regions from the extracted genome and the original sgRNA plasmid library, and Illumina sequencing was performed as described before[45]. sgRNA read count data were analyzed with MAGeCK (version 0.4.4)[25] and genes with enriched and depleted sgRNAs were detected using the test command (default parameters).

## Cas9 TNG MKOS MEF reprogramming

A total of $0.25 \times 10^4$ Cas9 TNG MKOS MEFs were mixed with $9.5 \times 10^4$ WT MEFs (129 strain) and seeded in gelatine-coated wells of 6-well plates. Cells were transduced with sgRNA lentiviruses at an MOI of 3 with 8 μg ml⁻¹ Polybrene (Merck-Millipore) for 4 h and then reprogramming was initiated by addition of reprogramming medium. On day 14–16, whole well colony images were taken using the Celigo S Cell Cytometer (Nexcelom) and the number of *Nanog*-GFP⁺ and *Nanog*-GFP⁻ colonies were counted. The images shown for illustration were stitched using Celigo S Cell Cytometer and processed using ImageJ.

## *piggyBac* reprogramming of MEFs with sgRNA expression and/or *Zfp266* cDNAs

*Nanog*-GFP MEFs with or without Cas9 expression from the *Rosa* locus isolated from E12.5 embryos, or wild type MEFs were plated at $1.5 \times 10^5$ cells per well in a gelatin-coated 6-well plate. Twenty-four h later co-transfection of a Dox-inducible *piggyBac* transposon vector carrying

the *tetO-MKOS-ires-mOrange* or *tetO-STEMCCA-ires-mOrange* cassette with sgRNA expression cassette, *PB-CA-rtTA* vector with/without carrying a *P2A*-linked *Zfp266* cDNAs, and *pCMV-hyPBase* was performed using 500 ng each DNA and 6 μl of FugeneHD (Promega) as per manufacturer's instructions[20,46,47]. Twenty-four h later reprogramming was initiated with reprogramming medium. Medium was changed every 2 days. For colony counting, whole well colony images were taken on day 14–16 using the Celigo S Cell Cytometer (Nexcelom) and colonies were counted with ImageJ.

## *piggyBac* reprogramming of NSCs with sgRNA expression

A GFP sgRNA vector was delivered into Cas9 and GFP expressing NSCs[48], kindly provided by the Pollard lab, University of Edinburgh, using nucleofection with the SG Cell Line 4DNucleofector X Kit (Lonza) as per manufacturer's instructions. GFP⁻ NSCs were sorted using the FACS AriaII (BD Biosciences) and plated at clonal density. Individual clones were picked and genotyped to confirm GFP KO. NSCs were reprogrammed by nucleofection of a Dox-inducible *piggyBac* transposon vector carrying the *tetO-MKOS-ires-mOrange* cassette with/without a sgRNA expression cassette, *PB-CA-rtTA* vector and *pCMV-hyPBase*. $2 \times 10^5$ NSCs were nucleofected with 750 ng each of the above-mentioned plasmids using SG Cell Line 4DNucleofector X Kit (Lonza), DN-100 program, as per manufacturer's instructions. Cells were recovered in NSC medium and then plated on a layer of wild type MEF feeder cells seeded the day before at a density of $1 \times 10^5$ cells per well in a gelatin-coated 6-well plate. One day post-nucleofection, reprogramming was initiated with NSC complete medium supplemented with 100 U ml⁻¹ human LIF, 0.3 μg ml⁻¹ of doxycycline (Sigma) and 10 μg ml⁻¹ of L-ascorbic acid or 2-Phospho-L-ascorbic acid trisodium salt (Sigma). After 6 days, the medium was switched to serum-free N2B27-based medium (containing DMEM/F12 medium supplemented with N2 combined 1:1 with Neurobasal® medium supplemented with B27; all from Thermo Fisher Scientific), MEK inhibitor (PD0325901, 0.8 μM, Axon Medchem), GSK3b inhibitor (CHIR99021, 3.3 μM, Axon Medchem), 1 μg ml⁻¹ of doxycycline (Sigma) and 10 μg ml⁻¹ of L-ascorbic acid or 2-Phospho-L-ascorbic acid trisodium salt (Sigma). At day 16 of reprogramming, immuno-fluorescence for NANOG was performed as follows: cells fixed with 4% paraformaldehyde for 10 minutes on day 14 were permeabilized in 0.1% Triton-X in PBS for 1 h, blocked in 5% BSA in PBS with 0.1% Tween20 for 1 h at room temperature, and then stained in blocking solution with a primary antibody for NANOG (eBioMLC-51, Thermo-fisher Scientific, Dilution: 1/1000) overnight at 4 °C. The next day, an AlexaFluor488 conjugated secondary antibody (A-21208, Invitrogen, Dilution: 1/1000) was applied in blocking solution for 45 minutes at room temperature before washing and imaging. Whole well images were taken using the Celigo S Cell Cytometer (Nexcelom) and colonies were counted with ImageJ.

## CD44, ICAM1, *Nanog*-GFP expression analysis during reprogramming

Cells harvested at different time points of reprogramming were stained in FACS buffer for 30 min at 4 °C and washed with FACS buffer (1% FCS in PBS) prior to acquisition with LSR Fortessa (BD Biosciences) cytometer. The following antibodies from eBioscience were used: ICAM1-biotin (13-0541-82; Dilution: 1/100), CD44-APC (17-0441-82; Dilution 1/300), streptavidin-PE-Cy7 (25-4317-82; Dilution: 1/1500). Dead cells were excluded using LIVE/DEAD™ Fixable Near-IR Dead Cell Stain Kit (ThermoFisher Scientific, Dilution: 1/1500). Data were analyzed using Flowjo v10.

## RNA-Seq

**Sample Preparation.** For Wt and *Zfp266* KO MEF samples, $1 \times 10^5$ Cas9 TNG MKOS MEFs were transduced with either a non-targeting control sgRNA or *Zfp266* sgRNA lentivirus at an MOI of 3 with 8 μg ml⁻¹

polybrene (Merck-Millipore) for 4 h. After additional 96 h culture in MEF media, the cells were harvested for RNA extraction. For reprogramming samples, $0.25 \times 10^4$ Cas9 TNG MKOS MEFs were mixed with $9.5 \times 10^4$ WT MEFs (129 strain) and seeded in gelatin-coated wells of 6-well plates. Cells were transduced with either a non-targeting control sgRNA or *Zfp266* sgRNA lentivirus at an MOI of 3 with 8 µg ml$^{-1}$ Polybrene (Merck-Millipore) for 4 h, before being recovered for 24 h in MEF media. After 24 h reprogramming was initiated by addition of reprogramming medium. Cells were harvested at day 3, day 5 and day 7 of reprogramming, respectively, and $1 \times 10^5$ of mOrange$^+$ OSKM expressing cells were sorted with the FACS AriaII (BD Biosciences) per sample. *Nanog*-GFP$^+$ iPSCs were harvested at day 15, and sorted with the FACS AriaII (BD Biosciences) into 96-well plates. Sorted iPSCs were cultured in ESC medium with puromycin (1 µg ml$^{-1}$) to select for transgene independent clones and KO of *Zfp266* was confirmed by genotyping. *Zfp266* KO ESCs were generated by transfecting Clone J ESCs with a *Zfp266* sgRNA plasmid expressing BFP. Single BFP$^+$ ESCs were then sorted with the FACS AriaII (BD Biosciences) into 96-well plates 48 h after transfection. Clones which became BFP- negative (i.e. shed the sgRNA plasmid) were selected and KO of *Zfp266* was confirmed by genotyping. $1 \times 10^5$ iPSCs or ESCs were used for RNA extraction. Cells were homogenized with the QIAshredder kit (Qiagen) and total RNA was extracted from all samples using the RNeasy Plus Micro Kit (Qiagen). Libraries were prepped with the NEB Ultra II stranded mRNA Library prep kit (NEB). RNA-Seq libraries were sequenced with NextSeq, 75SE.

**Read processing.** For each sequencing run, a quality control report was generated using FastQC and Illumina TruSeq adapter sequences were removed using Cutadapt[49]. Sequencing runs from the same biological sample were then concatenated and mapped to the GRCm38 reference genome using STAR[50].

**Differential analysis.** For each biological sample, aligned sequencing reads were first assigned to genomic features (e.g., genes) using Rsubread[51] and a count table was generated. Differential expression analysis was then performed with DESeq2[52], and statistically significant genes (e.g., FDR < 0.05 and log2FoldChange > 1) were identified using the standard workflow. Importantly, although the data represents a control and treatment time-series experiment, we opted to combine the factors of interest into a single factor for easier comprehension. Gene ontology analysis for differentially expressed genes was performed using the goseq package[53].

**Downstream analysis.** For exploratory analysis and visualization, a batch-corrected and regularized log matrix of expression values was used. The count table was first transformed to stabilize the variance across the mean using the rlog function from DESeq2 and then unwanted batch effects (e.g., library preparation date) were removed using the removeBatchEffect function from limma[54].

### DamID-seq
**Sample Preparation.** A total of $1 \times 10^5$ WT MEFs (129 strain) were nucleofected with either *PGK-mO-Dam* or *PGK-mO-Dam-Zfp266* plasmids using the P4 Primary Cell 4D-Nucleofector X Kit (Lonza). 5 replicates were performed in total. Cells were recovered in MEF media for 48 h before $3 \times 10^4 - 1.6 \times 10^5$ GFP + cells per sample were sorted with the FACS AriaII (BD Biosciences). Genomic DNA was isolated with Quick-gDNA™ MicroPrep (ZymoResearch) and 32 ng genomic DNA/sample was used for DamID-seq library preparation as previously described[37]. In brief, 32 ng genomic DNA was digested with 20 units of DpnI (NEB) for 3 h at 37 °C in 15 µl volume, before heat inactivation of the enzyme for 20 min at 80 °C. Double-strand DamID adapters were prepared by annealing AdRt (5'-CTAATACGACTCACTATAGGGCA GCGTGGTCGCGGCCGAGGA-3', IDT) and AdRb (5'-TCCTCGGCCG-3',

IDT)[38]. Adapter ligation was performed overnight at 16 °C with 20 units T4 ligase, 1× T4 ligase buffer (NEB), and 0.2 µM DamID adapters in 20 µL. After heat inactivation of ligase for 20 min at 65 °C, DpnII digestion was performed for 1 h at 37 °C before heat inactivation for 20 min at 65 °C. Then, 100 µL of PCR mix (KAPA HiFi HS ReadyMix (KAPA Biosystems), 10 µM AdR PCR primer, 5'-GGTCGCGGCCGAGGA TC-3' (IDT), 1× SYBR green I nucleic acid gel stain (Life Technologies)) was added to the tube. From the final volume of 125 µL, 10 µL was used to perform qPCR in technical duplicate (in total 20 µL), and the number of PCR cycles to stop the PCR in the log-linear amplification phase was determined. The remaining 105 µL was then used to amplify the adapter-ligated fragments. The amplified DNA fragments were purified using SPRI magnetic beads, and 50 ng of the PCR amplified DNA was used in a tagmentation reaction (5 min) in which the Tn5 enzyme fragments the DNA and simultaneously inserts the preloaded Illumina adapters. After PCR amplification of the tagmented DNA and SPRI magnetic beads purification, DamID libraries were sequenced with NextSeq, 40PE.

**Read processing.** For each sequencing run, a quality control report was generated using FastQC and Illumina Nextera adapter sequences were removed using Cutadapt. Sequencing runs from the same biological sample were then concatenated and mapped to the GRCm38 reference genome using BWA[55]. Uninformative and spurious alignments were subsequently filtered using a combination of SAMtools[56] and BEDtools[57] commands. Specifically, reads mapped to the mitochondrial chromosome and reads mapped to blacklisted regions were filtered.

**Peak calling.** For each biological sample, aligned sequencing reads were assigned to genomic features (e.g., DpnII restriction fragments) using Rsubread and a count table was generated. Statistically significant regions of Dam-fusion protein binding (e.g., FDR < 0.05 and log2FoldChange > 1) were detected using the callPeak command from Daim[37]. For further details, please refer to the original manuscript describing the Daim software[37]. The regions were then annotated and analyzed for gene and genome ontology enrichment using the annotatePeaks command from HOMER[58].

**Downstream analysis.** Heatmaps of read coverage at Dam-fusion binding regions were produced using the computeMatrix and plotHeatmap commands from deepTools[59]. When plotting heatmaps, a total of 5 peaks identified exactly over *Zfp266* exons (chr9:20495068-20521417) were removed from the *Zfp266* DamID peak regions due to the high signal intensity caused by the *PGK-mO-Dam-Zfp266* plasmid. De novo motif discovery and was performed using the MEME-ChIP tool from the MEME suite (version 5.1.1)[60]. Motif enrichment analysis was performed using findMotifsGenome command from HOMER[58] as DamID-seq's large peak size was not optimal for the MEME-ChIP tool. Genome browser images of peak regions and read coverage were composed using the Integrative Genomics Viewer[61].

### ATAC-seq
**Sample Preparation.** Cas9 TNG MKOS MEFs were plated and transduced in the same manner as samples prepared for RNA-Seq. After 24 h reprogramming was initiated by addition of reprogramming medium for reprogramming samples, while MEF samples were maintained in MEF media. Cells were harvested 96 h after sgRNA transduction (which was 72 h after OSKM induction for reprogramming samples) and sorted with the FACS AriaII (BD Biosciences). Cells were then processed for ATAC-Sequencing according to the Omni-ATAC protocol[62]. Briefly, $5 \times 10^4$ sorted MEFs or mOrange$^+$ OSKM expressing cells per sample were washed with cold 1x PBS then pelleted before the supernatant was discarded. Cell pellets were then gently resuspended in 50 µl of lysis buffer (48.5 µl resuspension buffer, 0.5 µl 10% NP-40 (Sigma)

0.5 µl 10% Tween-20 (Sigma), 0.5 µl 1% Digitonin (Promega) (resuspension buffer: 500 µl 1 M Tris-HCl, pH7.5 (ThermoFisher), 100 µl 5 M NaCl (Sigma), 150 µl 1 M MgCl$_2$ (Sigma), 49.25 ml nuclease-free H$_2$O) and incubated on ice for 3 minutes. Then, 1 ml of wash buffer (990 µl resuspension buffer, 10 µl Tween-20 (Sigma)) was added to the tubes before they were gently inverted and then centrifuged for 10 minutes at 500 x $g$, at 4 °C. Supernatants were then carefully aspirated. Nuclei pellets were then resuspended in 50 µl of transposition mix (2.5 µl Tn5 transposase, 25 µl 2x TD buffer (both Illumina), 0.5 µl 1% Digitonin (Promega), 0.5 µl 10% Tween-20 (Sigma), 16.5 µl 1x PBS, 5µl nuclease-free H$_2$O) and incubated in a thermomixer at 37 °C, 1000 rpm for 30 minutes. Transposed DNA was then purified with the Zymo DNA Clean and Concentrator-5 Kit (Zymo Research) and eluted in 21 µl nuclease-free H$_2$O. All purified DNA (~20 µl) was used for PCR amplification with NEBNext High Fidelity 2x MasterMix (NEB) and optimum cycle number was determined by qPCR, as per the protocol. Amplified DNA was then purified with double-sided bead purification using AMPure XP magnetic beads (Beckman Coulter). Library concentration was determined with Qubit (ThermoFisher) and fragment size/quality with TapeStation (Agilent). ATAC libraries were sequenced with NextSeq, 40PE.

**Read processing.** For each sequencing run, a quality control report was generated using FastQC and Illumina Nextera adapter sequences were removed using Cutadapt[49]. Sequencing runs from the same biological sample were then concatenated and mapped to the GRCm38 reference genome using BWA[55]. Duplicate reads caused by PCR amplification were subsequently identified using the MarkDuplicates command from Picard (https://broadinstitute.github.io/picard/). Uninformative and spurious alignments were next filtered using a combination of SAMtools[56] and BEDtools[57] commands. Specifically, reads mapped to the mitochondrial chromosome, reads mapped to blacklisted regions, reads marked as duplicates, and reads not properly paired (e.g., reads that aren't FR orientation or with an insert size greater than 2 kb) were filtered.

**Peak calling.** For each biological sample, statistically significant regions of chromatin accessibility (FDR < 0.1) were detected using the callpeak command from MACS2[63] (https://github.com/macs3-project/MACS). For downstream analyzes, a consensus set of peaks was created by taking the union across all biological samples with the multi-inter command from BEDtools[57].

**Differential analysis.** For each biological sample, aligned sequencing reads were first assigned to genomic features (e.g., consensus set of peaks) using Rsubread[51] and a count table was generated. Differential accessibility analysis was then performed with DESeq2[64] and statistically significant peaks (e.g., FDR < 0.05 and log2FoldChange > 1) were identified using the standard workflow.

**Downstream analysis.** Heatmaps of read coverage at chromatin accessibility regions were produced using the computeMatrix and plotHeatmap commands from deepTools[59]. K-means clustering was used to partition the regions into two distinct categories of reprogramming MORs. Genome browser images of peak regions and read coverage were composed using the Integrative Genomics Viewer[61]. Peaks were annotated against mm10 with annotatePeaks.pl from the HOMER suite (version 4.11)[58]. De novo motif discovery and enrichment analysis of MORs were performed using the *Zfp266* KO samples' narrowpeak summits within MORs with the MEME-ChIP tool from the MEME suite (version 5.1.1)[60]. The number of SINE elements around peaks were counted using the BEDTools window command in a window of ±500 bp from the summits of the peaks. ATAC-seq data of iPSCs were retrieved from GSE98124[39].

## ChIP-seq and MNase-seq data analysis

ChIP-seq data of ESCs and MEFs in early reprogramming at 48 h were retrieved from GSE90895 and GSE168142, respectively[40,41]. Read mapping and peak calling of the ChIP-seq data were performed with BWA and MACS2 as described above, as for the ATAC-seq data. Input control-subtracted coverage files were generated using the bdgcmp command from MACS2 and were transformed into bigWig using bedGraphToBigWig. Micrococcal nuclease-sequencing (MNase-seq) data of MEF was retrieved from GSE40896 in the BED format[65]. Coordinates for mm9 were converted to mm10 using the UCSC liftOver tool[66].

## Luciferase reporter assays

The pGL3 reporter plasmid containing the SV40 early promoter (Promega) was used for all luciferase reporter assays along with an internal control Renilla plasmid (Promega). Luciferase activity was measured with the GloMax 96-microplate luminometer (Promega) using the Dual-Glo Luciferase Assay System (Promega). For assays performed in HEK293 cells, 0.5–1 × 10$^4$ cells were plated per well of a 96-well plate 24 h prior to transfection. Transfection mixes were prepared as follows; 100 ng pGL3 reporter plasmid, 0.5 ng Renilla plasmid and 100 ng overexpression plasmid (BFP/VPR/VPR-*Zfp266*/Wt *Zfp266*) were mixed in Opti-MEM I Reduced Serum Medium (Gibco) up to 100 µl. Fugene HD Transfection Reagent (Promega) was then added at a ratio of 3:1 (reagent:DNA) and 5–10 µl was added to each well of cells. Luciferase activity was measured 48 h after transfection. For assays performed in MEF/reprogramming cells, 1 × 10$^4$ TNG MKOS MEFs were plated per well of a 96-well plate 24 h prior to transfection, either in MEF media or ES media +dox (300 ng ml$^{-1}$) to induce OSKM expression. Transfection mixes were prepared as such; 1 µg pGL3 reporter plasmid, 10 ng Renilla plasmid were mixed in Opti-MEM I Reduced Serum Medium up to 100 µl. Fugene HD Transfection Reagent was then added at a ratio of 4:1 (reagent:DNA) and 20 µl was added to each well of MEFs/reprogramming cells. Luciferase activity was measured 48 h after transfection.

## Reporting summary

Further information on research design is available in the Nature Portfolio Reporting Summary linked to this article.

## Data availability

RNA-seq, DamID-seq, ATAC-seq data generated in this study have been deposited in the GEO database under accession code GSE198166. The processed KO screen data are available at https://kaji-crispr-screen-updated.netlify.app. Source data are provided as a Source Data file. Source data are provided with this paper.

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

## Acknowledgements

We thank I. Chambers for providing TNG ESC line, F. Rossi and C. Cryer for assistance with flow cytometry, Biomed unit staff for mouse husbandry, the Wellcome Sanger Institute sequencing facility for gRNA sequencing, EMBL GeneCore for RNA-seq, ATAC-seq and DamID-seq, A. Soufi, D. O'Caroll and M.L. Huynh for comments on the manuscript. Some of the computations for this work were enabled by resources in project SNIC 2017/7-317 provided by the Swedish National Infrastructure for Computing (SNIC) at the Uppsala Multidisciplinary Center for Advanced Computational Science (UPPMAX). This work was supported by European Research Council (ERC) grants ROADTOIPS (no. 261075) and MRC senior non-clinical fellowship (MR/N008715/1) funded for K.K. We also thank the generous support from Baillie Gifford for the collaboration between CiRA and MRC CRM, from Japan Agency for Medical Research and Development (AMED) for CiRA. K.Y. was supported by the Wellcome Trust (206194). D.F.K., J.A. and M.Y. was supported by the BBSRC (EASTBIO doctoral training partnership), Principal's Career Development scholarship from the University of Edinburgh, and Japan Society for the Promotion of Science (JSPS) Overseas Research Fellowships, respectively. V.O. and M.Bertenstam were supported by the Swedish Foundation for Strategic Research (A3 04 159p). V.O. was also supported by the Swedish Research Council (Vr 621-2008-3074). S.K. was supported by Jane and Aatos Erkko Foundation.

## Author contributions

D.F.K. designed and performed sgRNA screen, validation and characterization of the roadblock genes including *Zfp266*. M.Y., J.A., S.K. and S.R.T. contributed to the analyzes of the gRNA sequencing, RNA-seq, ATAC-seq, ChIP-seq and DamID-seq data sets. M.B., S.Z. and K.O. provided technical support. M.Bertenstam and V.O. generated the screening data website. K.Y. provided the gRNA library, the Rosa26-Cas9 targeting vector, and advised on the screen. K.K. conceived the study, supervised experiment design and data interpretation, and wrote the manuscript with D.F.K.

## Competing interests

The authors declare no competing interests.
