## [Peer Review File · Nature Communications]

B1 SINE-binding ZFP266 impedes mouse iPSC generation through suppression of chromatin opening mediated by reprogramming factorsReviewer #1 (Remarks to the Author):

Kaemena et al report a new reprogramming roadblock, ZFP266, identified from a genome wide CRISPR LOF screen. ZFP266 a C2H2 zinc finger with a KRAB suppressor domain which hasn't been implicated in reprogramming – in fact I couldn't find any pubmed entry for this gene. Depletion of ZFP266 increases reprogramming efficiency. They show that ZFP266 binds the B1 SINE group and propose a model whereby this binding brings ZFP266 close to functionally important OSK sites which mitigates their binding and chromatin opening. Removing ZFP266 grants easier access of OSK to their targets and facilitates reprogramming. Interchanging the KRAB domain with VPR turns ZFP266 into a facilitator of reprogramming. Findings are interesting and shows ones more that iPSC induction combined with genome wide screens presents a powerful system to study basic mechanisms of cellular fate decision, reveals the importance of transposable elements to shape cistrome and gene expression. It uncovers a novel role of a KRAB zinc fingers in preserving cell states. Experiments are overall technically sound and presented in a pleasant manner - I am supportive to publish this work in NC after revision. I would however like to see some more elaboration as to the cross-talk of ZFP266 with OSK and compacted chromatin. A closer look at available data might allow them to achieve this.

Major points:

1) If I understand correctly the model (Figure 6O) posits that ZFP266 binds B1 LINE derived DNA sites that are nucleosome free. Then the KRAB domain extends and keeps chromatin encoding OSK binding sites compacted (OSK sites are ~70 bp away from ZFP266/B1 sites). I note that ZFP266 is composed of 11 C2H2 zinc fingers. KLF4 was proposed to use only two of its three ZNFs to bind nucleosomes as all three couldn't wrap around the DNA in the context of nucleosomal DNA. So I presume authors propose the actual binding sites of ZFP266 is NCP free but the nearby OSK sites are not? If authors propose that ZFP266 itself might directly bind NCPs to keep them compacted some further discussion and analysis would be appreciated. ATAC-seq and focused motif analysis should in principle allow to determine TF footprints to clarify whether this is indeed the case (a number of tools are available for such analysis). Biochemical assays (using re-constituted chromatin/NCPs and purified TFs) might be beyond the scope of the present work but a high resolution ATAC-seq analysis might help to back up this model. If KRAB domains maintain chromatin compaction and prevent access and loosening by "pioneer" TFs that would be a conceptually very interesting insight. What is the significance of the 70bp distance between SINES and MOR summits? Are the summits at an NCP dyad and SINES and entry/exit sites? Does the binding of ZFP266 to B1 with a 70bp spacing have some role in NCP positioning?

2) For a novel factor it maybe be difficult to verify the KO on a protein level. Yet, expression levels of ZFP266 throughout reprogramming should be shown for WT and KO situation. It seems ZFP266 has the expression pattern of a housekeeping gene. In Supplemental Table S3 and S4, compared with WT cells, Zfp266-KO cells show similar read counts – please explain. Were ZFP226 mRNA levels analysed by qPCR ?

3) Some additional basic characterisation of ZFP266 KO iPSCs would be useful. Figure 3G suggest KO iPSCs have global genes expression as WT iPSCs/ESCs. Are KO iPSCs compromised in their pluripotency and self-renewal? I wouldn't insist on in vivo characterisation but some basic in vitro QC of ZFP266 KO iPSC lines are recommended (differentiation, passaging, karyotype).

Minor points:

1) Authors may considered using the C2H2 recognition code to predict which finger bind to which nucleotides of the de novo pwms. Sangamo uses this code to design genome editing therapeutics. If authors have access to these techniques biochemical assays to verify specific of ZFP266 to B1 derived DNA would be of interest.

2) The genome-wide screening is performed on Cas9-expressing MEFs. And the KO effects of those identified roadblock genes are also examined in Cas9-expressing MEFs and NSCs. Does Zfp266-KO show similar effect in iPSC generation using starting cells

from different genetic background (Oct4-GFP, Sox2-GFP or WT MEFs, fibs from adult mice)?

3) I find figure 5I very interesting. It seems variants of the POU motifs (SoxOct versus octamer have different patterns). KLF and SOX motifs appear to be enriched at a periodic distance to the summit. Any reason for this (Jussi Taipales NCP SELEX analysis might help). I wonder if the pattern would differ of full KLF and the shortened KLF motif (both motifs feature in Figure S5f).

4) Authors may wish to more explicitly name the various POU motifs they find (octamer, MORE, canonical SoxOct)

5) Methods are at times too brief and excessively refer to previous work.

6) Was DamID performed in the context of ZFP266 KO or WT MEFs (where endo ZFP222 is still present).

Reviewer #2 (Remarks to the Author):

This study uses an unbiased screen to identify new inhibitors of iPSC conversion, using a previously established screening and analysis platform. Overall, the results are quite convincing in proposing an important role for Zfp266 in blocking iPSC conversion, and the data are very well presented. Many controls and parallel approaches are used to support their main arguments and conclusions, and for the biggest part, most of my concerns/criticism that arose while reading the story were taken away by the next paragraph and figure. So overall, the data and its conclusions appear solid and would be a great contribution to the field of iPSC conversion. There are however a few things that should be taken into consideration and addressed before this article is suitable for publication.

Main points

The authors identify Zfp266 as a mouse roadblock protein to iPSC reprogramming. Obviously, establishing conditions that improve reprogramming of iPSCs is particularly important for human iPSCs generation, with its many possible pre-clinical and clinical applications. One big limitation of the current study is that even though the results of mouse Zfp266 are quite convincing, zfp266 is actually a murine (mouse)-specific ZNF gene and protein. After checking this by BLAT searches on protein-sequence level, it turns out that it does not have a direct or even closely related ortholog in humans. Because its role in blocking reprogramming in mouse cannot be extrapolated to other species and therefore its usability for human research is therefore very limited. The authors should mention more clearly in the abstract and introduction/discussion that the identified inhibitors of iPSC conversion are found in mouse, and clearly state that Zfp266 is a gene that is not present in humans (although other KZNFs in humans may have evolved to get a similar function as mouse Zfp266)

In Figure 1E, mOrange represents the presence of the MKOS factors/construct used for reprogramming, and green represents the presence of the pluripotency factor NANOG. What is curious from these pictures is that KO of both known reprogramming inhibitors TRP53 and Cdkn1a show a high level of mOrange, just like the novel factor FAM122a, but this is not the case for Zfp266: Is there a possibility that Zfp266 directly induces NANOG in cells that have not well converted to pluripotency (due to low levels of MKOS/mOrange)? Even though my concerns about this were largely taken away by the later figures/analyses, it may be important for the reader that the authors take away that concern, also regarding the data in Figure 2, possibly by including Flow sorting data that confirms the consistent presence of mOrange (and therefore the reprogramming factors) in the resulting colonies, because this is not entirely clear from these pictures.

Figure 2: Continuing on my foregoing question: The authors should provide a (possible)

explanation for the abundance of mOrange+ / Nanog- cells in the two most efficient known reprogramming inhibitors TRP53 and Cdkn1a? Again, Zfp266 has a qualitatively different appearance in the alternative assays shown here. The authors should address this in the text and perhaps add data to convince that reprogramming is occurring at the same level/with the same dynamics as the previously described reprogramming inhibitors under Zfp266 KO conditions (see the point above). Again, FAM122a seems a very interesting target to follow up as well, given it's similarities in appearance to TRP53 and Cdkn1a, but it's not addressed as such in the text. Is there a reason for this?

Figure 3: Why are the cells not followed up/analysed at later stages? The PCR plot shows that d7 cells during conversion are clearly different from MEFs, but they are still a long way from iPSCs (with fulfilled conversion). It would be important to show that the zfp266 KO cells can fully convert to iPSCs.

Figure 5; Line 221: the overlap between ATAC-seq peaks and DamID-seq was very limited, probably because the DamID-seq was performed in MEFs, and the ATAC-seq 72 hours into iPSC conversion. Although the data are quite convincing, it would be a nice addition to show Zfp266 binding sites at the 72 hour timepoint to explain this discrepancy and get more support for a direct link between the ATAC-seq data and actual Zfp266 binding

Minor points

- it is unclear in the title, abstract and introduction in which species the analysis is done. There should be a clear mentioning of the species.
- The last paragraph of the intro reads much like an abstract and should be changed.
- Have the authors considered making double or triple KOs of the different pluripotency blockers to see if there is an additive effect?
- BFP is not explained anywhere in the manuscript. I assume Blue Fluorescent Protein?
- In figure 3D, I feel it's better to indicate 'wt-zfp266' instead of 'sgRNA-resistant'

Reviewer #3 (Remarks to the Author):

This manuscript from Kaemena et al reports that the KRAB zinc finger protein (KRAB-ZFP) Zfp266 modulates somatic reprogramming through inhibiting the chromatin remodeling mediated by pioneering factors. They applied genome-wide CRISPR KO screening and identified Zfp266 as a novel repressor for somatic reprogramming. Knockout of Zfp266 improved reprogramming efficiency while overexpression (OE) of Zfp266 impedes reprogramming. By using DamID-seq and ATAC-seq, the authors observed that ZFP266 binds SINE-B1 elements adjacent to Yamanaka factors Oct4/Sox2/Klf4 binding sites and then suppresses chromatin opening mediated by these pioneer factors.

However, many of the data are over-interpreted, which not fully support their statements. For example, there is no direct evidence that SINE-B1 elements regulate the genes suggested by the authors, and it lacks function validations for those genes. This manuscript also lacks the biological insight in this field, thus I would suggest considering the manuscript more suitable for another journal.

Major points

1. In Figure4, there are 15119 ZFP266 binding sites, but only 123 of which are MORs after Zfp266-KO, indicating that the major function of ZFP266 is still unknown instead of

suppression chromatin accessibility as the authors claim.

2. The Luciferase reporter assay cannot represent the effects of endogenous genomic elements faithfully. For Figure 6E, the authors claim that SINE act as regulatory elements for the *B3gnt3*, *Piwil2*, *Snx20*. It is required to delete those SINE-loci and to check if such deletions impair the expression of respective gene. A SINE-B1 element is very short, allowing easy validation by CRISPR-KO.

3. Figure 6E, the ChIP-seq data signal seems too sparse to be real binding signal, especially at the *B3gnt3* loci. Input control is required to exclude any biased interpretation.

4. The mechanism of *Zfp266*-KO facilitates reprogramming is still unknown. The authors have listed a few genes, but whose function seems still unclear. Can overexpression of one or more these genes promote reprogramming?

RESPONSE to REVIEWER COMMENTS

Reviewer #1 (Remarks to the Author):

Kaemena et al report a new reprogramming roadblock, ZFP266, identified from a genome wide CRISPR LOF screen. ZFP266 a C2H2 zinc finger with a KRAB suppressor domain which hasn't been implicated in reprogramming – in fact I couldn't find any pubmed entry for this gene. Depletion of ZFP266 increases reprogramming efficiency. They show that ZFP266 binds the B1 SINE group and propose a model whereby this binding brings ZFP266 close to functionally important OSK sites which mitigates their binding and chromatin opening. Removing ZFP266 grants easier access of OSK to their targets and facilitates reprogramming. Interchanging the KRAB domain with VPR turns ZFP266 into a facilitator of reprogramming. Finding are interesting and shows ones more that iPSC induction combined with genome wide screens presents a powerful system to study basic mechanisms of cellular fate decision, reveals the importance of transposable elements to shape cistrome and gene expression. It uncovers a novel role of a KRAB zinc fingers in preserving cell states. Experiments are overall technically sound and presented in a pleasant manner - I am supportive to publish this work in NC after revision. I would however like to see some more elaboration as to the cross-talk of ZFP266 with OSK and compacted chromatin. A closer look at available data might allow them to achieve this.

We very much appreciate this reviewer's support.

Major points:

1) If I understand correctly the model (Figure 6O) posits that ZFP266 binds B1 LINE derived DNA sites that are nucleosome free. Then the KRAB domain extends and keeps chromatin encoding OSK binding sites compacted (OSK sites are ~70 bp away from ZFP266/B1 sites).

I note that ZFP266 is composed of 11 C2H2 zinc fingers. KLF4 was proposed to use only two of its three ZNFs to bind nucleosomes as all three couldn't wrap around the DNA in the context of nucleosomal DNA. So I presume authors propose the actual binding sites of ZFP266 is NCP free but the nearby OSK sites are not? If authors propose that ZFP266 itself might directly bind NCPs to keep them compacted some further discussion and analysis would be appreciated. ATAC-seq and focused motif analysis should in principle allow to determine TF footprints to clarify whether this is indeed the case (a number of tools are available for such analysis). Biochemical assays (using re-constituted chromatin/NCPs and purified TFs) might be beyond the scope of the present work but a high resolution ATAC-seq analysis might help to back up this model. If KRAB domains maintain chromatin compaction and prevent access and loosening by "pioneer" TFs that would be a conceptually very interesting insight. What is the significance of the 70bp distance between SINEs and MOR summits? Are the summits at an NCP dyad and SINEs and entry/exit sites? Does the binding of ZFP266 to B1 with a 70bp spacing have some role in NCP positioning?

We thank to this valuable comment. We were also interested in the meaning of the 70 bp distance between the midpoint of SINE and MOR summits.

Unfortunately DamID-seq peak summits do not represent Zfp266 binding sites as DamID-seq peaks depend on GATC sequence positions in the genome. However, knowing that ZFP266 binds B1 SINEs in reporter assays (Figure 6), we have looked into the relationship among KLF/SOX motifs, B1 SINEs within reprogramming MORs and nucleosome position.

First, we found that KLF and SOX motifs are enriched in the head and the tail of B1 SINEs, respectively (Figure 5K). It is probably due to G, A rich sequences in the head and the tail of B1 SINEs, making it easy to generate KLF and SOX motifs via small mutations. The size of SINEs is ~140 bp, thus this explains why the distance between the SINE midpoint and MOR summits/KLF, SOX motifs is 70 bp.

Second, we confirmed that B1 SINEs in the reprogramming MORs are enriched in nucleosome dyad in MEFs, with KLF/SOX motif enrichment (particularly KLF motif) at the nucleosome DNA entry-exit sites (Figure 5L). In fact, this feature was not specific to SINEs within MORs, and the same could be observed using all B1 SINEs while enrichment of KLF4 was less prominent (Supplementary Figure S6F). Notably, primate-specific Alu elements, which is closely related to rodent B1 SINEs, is known to have strong

association with nucleosomes (*Nat. Commun.* 11, 1–13 (2020)), thus it is likely due to DNA sequence of B1 SINE and Alu.

Based on these data, we propose the revised model (Figure 60), in which KLF4 and SOX2 bind to their motifs at the nucleosome DNA entry-exit sites at B1 SINE-associated nucleosomes, where ZFP266 impedes chromatin opening.

We appreciate this reviewer’s comment that allowed us to make this finding and propose the more accurate model.

Figure 5K. Numbers of SINEs located on the plus (1, 3) or minus (2, 4) strand either upstream (1, 2) or downstream (3, 4) of the MOR summits in reprogramming MORs (left top) and KLF4/SOX3 motif enrichment in each group with midpoint of B1 SINE in the centre (right four panels).

Figures 5L. Nucleosome dyad frequency at B1 SINEs within reprogramming MORs using MNase-seq data with MEFs (GSM1004654) (top) and KLF4 and SOX3 motif enrichment at the same regions (bottom) (5L). **Figure 5M.** A model of nucleosome wrapped by B1 SINE with KLF and SOX motifs at the head and tail.

Supplementary Figures S6F. Nucleosome dyad frequency at all B1 SINEs using MNase-seq data with MEFs (GSM1004654) (top), and KLF4 and SOX3 motif enrichment at the same regions (bottom).

Figures 60. Mechanistic model of how Zfp266 KO enhances reprogramming. B1 SINEs are strongly associated with nucleosome and often have KLF and SOX motif at the head and tail due to G and A rich sequence, respectively. ZFP266 binds to some of SINEs in a cell type- and context-dependent manner. Upon reprogramming factor expression, ZFP266 recruited to B1 SINE with KLF/SOX motifs at its head or tail, respectively, impedes chromatin opening (top). Zfp266 KO results in increased chromatin accessibility in those loci, facilitating pluripotency gene expression.

2) For a novel factor it maybe be difficult to verify the KO on a protein level. Yet, expression levels of ZFP266 throughout reprogramming should be shown for WT and KO situation. It seems ZFP266 has the expression pattern of a housekeeping gene. In Supplemental Table S3 and S4, compared with WT cells, Zfp266-KO cells show similar read counts – please explain. Were ZFP226 mRNA levels analysed by qPCR?

mRNA expression pattern of *Zfp266* during reprogramming is shown in Figure S2B. This reviewer's point is correct. We did not observe a difference in *Zfp266* mRNA expression between WT and *Zfp266* KO samples

Supplementary Figure S4A. Indel mutation frequency estimated by a TIDE assay with *Zfp266* cDNA from Cas9 TNG MKOS MEFs with *Zfp266* sgRNA expression used for RNA-seq (*Zfp266* KO MEF samples).

(Response Figure R1). However, we have performed TIDE assay (<https://academic.oup.com/nar/article/42/22/e168/2411890>), which evaluates insertion deletion (Indel) frequency, using *Zfp266* cDNA from a portion of MEFs prepared for RNA-seq samples (added as Supplementary Figure S4A). In all the 3 replicates, we could see more than 80% of out-of-frame indel in exon 6, and we confirmed that all the major indels cause premature stop codon >100 bp up-stream of last exon-exon junction. Thus, we believe that the maintained mRNA expression in *Zfp266* KO samples is due to inefficient nonsense-mediated RNA decay (NMD) against this mRNA but functional protein is not produced. However, we cannot confirm it because of the unavailability of antibodies against ZFP266.

Zfp266 mRNA in WT and *Zfp266* KO samples in the RNA-seq data.

3) Some additional basic characterisation of ZFP266 KO iPSCs would be useful. Figure 3G suggest KO iPSCs have global genes expression as WT iPSCs/ESCs. Are KO iPSCs compromised in their pluripotency and self-renewal? I wouldn't insist on in vivo characterisation but some basic in vitro QC of ZFP266 KO iPSC lines are recommended (differentiation, passaging, karyotype).

We thank these suggestions and added new figures as Supplementary Figure S4B-H. In short, *Zfp266* KO iPSCs had similar pluripotency gene expression and proliferation rates as WT iPSCs (Supplementary Figures S4B, S4C). They could down-regulate pluripotency genes and up-regulate 3 germ layer markers upon embryoid body differentiation (Supplementary Figures S4D).

Supplementary Figures S4B. Pluripotency gene expression extracted from RNA-seq data of iPSC clones established from *Zfp266* KO vs Wt reprogramming. **Supplementary Figures S3C.** Proliferation of Wt and *Zfp266* KO iPSCs over 15 days. **Supplementary Figures S4D.** Expression of pluripotency (Nanog, *Zfp42*, *Dppa5a*), ectoderm (Nestin), mesoderm (*Fgf8*, *Hand1*), and endoderm (*Gata4*, *Gata6*) markers upon embryoid body differentiation of Wt and *Zfp266* KO iPSCs.

However, we could identify 956 differentially expressed genes (DEGs) between iPSC clones generated by *Zfp266* KO and WT reprogramming (FDR<0.05, log₂FC >|1|) (Supplemental Figure S4E). It was in contrast to a relatively small number of DEGs (206, FDR<0.05, log₂FC >|1|) between *Zfp266* KO and wild type ESCs (Supplemental Figure S4F). GO term analysis for upregulated genes in both *Zfp266* KO ESCs/iPSCs showed enrichment for meiotic genes, while KO iPSCs further returned terms related to germ cell, reproduction and gamete generation. Downregulated terms in *Zfp266* KO relate to general development/neural development

Supplementary Figures S4E and S4F. RNA-Seq volcano plot of iPSC clones established from *Zfp266* KO vs Wt reprogramming (E) and *Zfp266* KO vs Wt ESCs (F). Up-regulated and down-regulated genes in KO cells are shown to the right and left of the plot, respectively (cut-off FDR<0.05, log₂FC >|1|). **Supplementary Figures S4G and S4H.** GO enrichment analysis of differentially expressed genes in *Zfp266* KO vs Wt iPSCs (G) and ESCs (H).

while ESC terms relate to blood lineage development (Supplemental Figure S4G and S4H). We believe these differences in gene expression in iPSCs are due to aberrant chromatin accessibility increase at some loci during reprogramming, highlighting the importance of Zfp266/B1 SINE-mediated gene expression control. This information is integrated in page 8-9.

Minor points:

1) Authors may consider using the C2H2 recognition code to predict which finger binds to which nucleotides of the de novo PWMs. Sangamo uses this code to design genome editing therapeutics. If authors have access to these techniques/biochemical assays to verify specific ZFP266 to B1 derived DNA would be of interest.

A cluster of 3-4 sequentially close ZF domains is usually required for specific DNA binding, and the selection of sequential 3 ZFs in the PWM predictor generates predicted DNA binding sequences of 10 bp. None of the 10 bp windows closely matched with a part of the B1 SINE consensus sequence. Thus, it is possible that ZFP266 recognizes a part of B1 SINE sequence with non-sequential ZFs. If it is the case, the number of the combinations with 3 ZFs are 84 ($9 \times 8 \times 7 / 3! / 1$). We have checked all the 84 combinations, but again none of the predicted DNA binding sequences matched well with the B1 SINE consensus sequence. The prediction might not be correct or a (multiple) partial matching is sufficient for binding. Either way, we could not identify a plausible binding sequence with the PWM predictor.

In order to find DNA binding sequence, we could try to identify essential ZFs for B1 SINE binding by deleting ZFs one by one, and then only use the essential ZFs in the PWM prediction tool. This could be followed by mutation in the closest sequences in B1 SINE to identify the ZFP266 binding sequence. It is certainly an interesting project, but we do think that it is beyond the scope of our current manuscript.

2) The genome-wide screening is performed on Cas9-expressing MEFs. And the KO effects of those identified roadblock genes are also examined in Cas9-expressing MEFs and NSCs. Does Zfp266-KO show similar effect in iPSC generation using starting cells from different genetic background (Oct4-GFP, Sox2-GFP or WT MEFs, fibs from adult mice)?

We are not totally sure if this question is referring to the genetic background, different reporters or different starting cell types. But the genetic background of our Cas9 expressing reprogrammable MEFs is pure 129. MEFs used for piggyBac reprogramming had mixed genetic background of 129, C57BL/6 and CD1. NSCs used in this work were derived from C57BL/6 mice. In all the cases, Zfp266 KO enhanced iPSC generation.

3) I find figure 5I very interesting. It seems variants of the POU motifs (SoxOct versus octamer have different patterns). KLF and SOX motifs appear to be enriched at a periodic distance to the summit. Any reason for this (Jussi Taipales NCP SELEX analysis might help). I wonder if the pattern would differ of full KLF and the shortened KLF motif (both motifs feature in Figure S5f).

We think that the difference between OCT and OCT4::SOX2 motif enrichment is due to SOX2 motif being rich in A and tends to be enriched in the tail of B1 SINE, as we described above. Periodic appearance of KLF, SOX motifs might also be due to the sequences within B1 SINE, as motif enrichment patterns within SINE in Figure 5K are closely associated with the orientation of B1 SINE.

As for full and short KLF4 motif enrichment, the enrichment pattern was similar between full and short KLF4 motifs as shown below (Response Figure R2), which is not surprising as the short motif is a part of the long motif.

Response Figure R2. Full and short Klf4 motif enrichment analysis at Zfp266 KO reprogramming MORs.

4) Authors may wish to more explicitly name the various POU motifs they find (octamer, MORE, canonical SoxOct)

We found smaller enrichment of POU motifs, compared to KLF, SOX, OCT4::SOX2 motifs, and did not find enrichment of MORE motif in reprogramming MORs. We believe it is because reprogramming MORs are generated due to KLF4/SOX2 binding to KLF, SOX motifs in the head and the tail of B1 SINE.

5) Methods are at times too brief and excessively refer to previous work.

We agreed that the method of DamID-seq sample preparation was too brief, thus added more detail. We also added how the published ChIP-seq data sets were analysed. If anything else should be added, please let us know. We are happy to do so.

6) Was DamID performed in the context of ZFP266 KO or WT MEFs (where endo ZFP222 is still present).

It was in the presence of endogenous ZFP266, as in a standard DamID-seq (Tosti, et al., 2018)

Reviewer #2 (Remarks to the Author):

This study uses an unbiased screen to identify new inhibitors of iPSC conversion, using a previously established screening and analysis platform. Overall, the results are quite convincing in proposing an important role for Zfp266 in blocking iPSC conversion, and the data are very well presented. Many controls and parallel approaches are used to support their main arguments and conclusions, and for the biggest part, most of my concerns/criticism that arose while reading the story were taken away by the next paragraph and figure. So Overall, the data and its conclusions appear solid and would be a great contribution to the field of iPSC conversion.

We thank this reviewer's positive evaluation and constructive suggestions.

Main points

The authors identify Zfp266 as a mouse roadblock protein to iPSC reprogramming. Obviously, establishing conditions that improve reprogramming of iPSCs is particularly important for human iPSCs generation, with its many possible pre-clinical and clinical applications. One big limitation of the current study is that even though the results of mouse Zfp266 are quite convincing, zfp266 is actually a murine (mouse)-specific ZNF gene and protein. After checking this by BLAT searches on protein-sequence level, it turns out that it does not have a direct or even closely related ortholog in humans. Because its role in blocking reprogramming in mouse cannot be extrapolated to other species and therefore its usability for human research is therefore very limited. The authors should mention more clearly in the abstract and introduction/discussion that the identified inhibitors of iPSC conversion are found in mouse, and clearly state that Zfp266 is a gene that is not present in humans (although other KZNFs in humans may have evolved to get a similar function as mouse Zfp266)

We appreciate this point, and have now stated that ZFP266 is an inhibitor of mouse iPSC generation in the abstract and introduction.

Regarding human, we found ZNF266, a potential orthologue protein of mouse ZFP266, has relatively low similarity, but 2 other KRAB-ZFPs, ZNF561 and ZNF562, have higher sequence similarities (Response Figures R3A and R3B), with ZNF561 having ~70% homology at the residues that specify target sequence (amino acids at positions -1, 2, 3, 6 within the ZF). Among those, only the overexpression of ZNF561 led to the inhibition of mouse iPSC generation in the absence of endogenous Zfp266 in a KRAB domain-dependent manner (Response Figures R3C-R3F), indicating that ZNF561 is likely an orthologue of mouse ZFP266. Mouse B1 SINEs are closely related to Alu elements in human genome, which have 2 7SL RNA-like sequences (Response Figure R3G). When the B1 SINE within the Snx20 reporter was replaced by Alu, VPR-Zfp266 could induce the reporter expression (Response Figure R3H), suggesting that ZFP266, and probably ZNF561, can bind to Alu.

Nevertheless, VPR-Zfp266 did not enhance human iPSC generation by OSKM (Response Figure R3I).

Interestingly, only 5.9% of ATAC-seq peaks in human ESCs (9,210/155,757) have Alu (Response Figure R3J), although KLF and SOX family motif enrichment were observed near the beginning of the 2nd 7SL-like sequence and at the end of these Alu within hESC ATAC-seq peaks, respectively (Response Figure R3K). In contrast, >20% of ATAC-seq peaks in mouse iPSCs (39,353/178,112) have B1 SINE, many of which are associated with pluripotency genes (Supplementary Figure S7).

We speculate this non-conserved positions of B1 SINE and Alu elements in the genome is at least in part the reason of VPR-Zfp266 not being able to enhance human iPSC generation.

We did not include the above mentioned human data in the manuscript, but if the reviewers and editors think that they could be good additional information, we would be happy to include them in the manuscript as supplementary figures.

Response Figure R3A. Amino acid sequence alignment of mouse ZFP266, human ZNF266, ZNF561 and ZNF562. **Response Figure R3B.** Neighbour joining Tree of mouse ZFP266, human ZNF266, ZNF561 and ZNF562. **Response Figure R3C and R3D.** *Cas9 Nanog*-GFP MEF reprogramming with MKOS piggyBac transposons, *Zfp266* sgRNA expression as well as cDNA overexpression of BFP, wild-type *Zfp266* (Wt), sgRNA resistant *Zfp266* (*gRNA Res*), sgRNA resistant *Zfp266* with KRAB domain deletion (Δ KRAB+L), human *ZNF266*, *ZNF561* or *ZNF562*, imaged at day 15. Red; mOrange, Green; Nanog-GFP (R3C). Mean Nanog-GFP⁺ colony numbers of C (R3D). Error bars indicate SEM, *p<0.05, **p<0.01, ***p<0.001 based on a one-way ANOVA test. **Response Figure R3E and R3F.** *Cas9 Nanog*-GFP MEF reprogramming with MKOS piggyBac transposons, *Zfp266* sgRNA expression as well as cDNA overexpression of BFP, sgRNA resistant *Zfp266* (*gRNA Res*), sgRNA resistant *Zfp266* with KRAB domain deletion (Δ KRAB+L), human *ZNF266*, *ZNF561* or *ZNF562* with (Δ KRAB+L) or without (Wt) KRAB domain deletion, imaged at day 15. Red; mOrange, Green; Nanog-GFP (R3E). Mean Nanog-GFP⁺ colony numbers of C (R3F). Error bars indicate SEM, **p<0.01 based on a one-way ANOVA test. **Response Figure R3G.** DNA sequence alignment of mouse B1 SINE and human Alu. **Response Figure R3H.** Luciferase reporter assay with an empty reporter vector (pGL3) or vectors containing *Snx20* regulatory region (*Snx20* R), *Snx20* regulatory region with B1 SINE deletion (*Snx20* R Δ B1) and *Snx20* regulatory region where B1 SINE as substituted with an Alu sequence (*Snx20* R Δ B1+Alu), co-transfected with either BFP, VPR only or VPR-*Zfp266* expression vectors in HEK293 cells. Error bars represent SEM of three biological replicates. **Response Figure R3I.** Human dermal fibroblast reprogramming using lentiviral vectors expressing *OCT4*, *SOX2*, *KLF4* and *MYC*, with VPR-*ZFP266*. NANOG⁺ colony numbers were counted on day 21 of reprogramming. Graph represents averages of 2 technical replicates. Error bars indicate standard deviation (s.d.). **Response Figure R3J.** Heatmap of human ESC ATAC-seq peak regions with (top) or without (bottom) overlap with Alu. **Response Figure R3K.** KLF4 and SOX3 motif enrichment at Alu within human ESC ATAC-seq peaks. Alu on the top and bottom strand are analysed separately.

Supplementary Figures S7. B1 SINEs in iPSC open chromatin regions. **A.** Of 178,112 iPSC ATAC-peaks, 22% (39,353) has at least one B1 SINE, majority of which are closed in MEFs. **B.** Those B1 SINEs at iPSC ATAC-peaks have KLF4 and SOX3 motif enrichment at the head and the tail of SINE. **C.** Gene associated with 12,150 iPSC ATAC-seq peaks with at least one B1 SINE and KLF/SOX motif are enriched in genes with a GO term “stem cell population maintenance”. Well known pluripotency associated genes are highlighted in magenta. **D.** Pluripotency gene loci with iPSC ATAC-seq peaks (red), B1 SINE (green) and KLF4 ChIP-seq signals (Blue). B1 SINE with KLF4 motif is indicated in dark green, and proximity of B1 SINEs to KLF4 ChIP-seq peaks is indicated in red arrows.

In Figure 1E, mOrange represents the presence of the MKOS factors/construct used for reprogramming, and green represents the presence of the pluripotency factor NANOG. What is curious from these pictures is that KO of both known reprogramming inhibitors TRP53 and Cdkn1a show a high level of mOrange, just like the novel factor FAM122a, but this is not the case for *Zfp266*: Is there a possibility that *Zfp266* directly induces NANOG in cells that have not well converted to pluripotency (due to low levels of MKOS/mOrange)? Even

though my concerns about this were largely taken away by the later figures/analyses, it may be important for the reader that the authors take away that concern, also regarding the data in Figure 2, possibly by including Flow sorting data that confirms the consistent presence of mOrange (and therefore the reprogramming factors) in the resulting colonies, because this is not entirely clear from these pictures.

We and others have previously shown that down-regulation of exogenous Yamanaka factors correlates with up-regulation of pluripotency genes at the late phase of successful iPSC generation. In our reprogramming system, mOrange^{high} cells hardly overlap with Nanog-GFP⁺ cells as below, thus we can consider mOrange^{high} cells are partially reprogrammed cells (Response Figures R4A and R4B) (O'Malley, et al., Nature, 2013).

Trp53 and *Cdkn1a* KO suppress reprogramming factor-induced senescence and apoptosis (Banito, Genes Dev. (2009), Hong, Nature (2009), Kawamura, Nature (2009), Li, Nature (2009), Marión, Nature (2009), Utikal, Nature (2009)), increasing the number of *Nanog*-GFP⁺ iPSC colonies, as well as partially reprogrammed *Nanog*-GFP⁺/mOrange^{high} colonies (Figure 1E, 1F, 2A, 2B, 2E, 2F). This reviewer is correct that *Fam122a* KO shows a similar phenotype as *Trp53* KO, indicating that *Fam122a* KO increases *Nanog*-GFP⁺ colony numbers via suppressing reprogramming factor-induced senescence and/or apoptosis similarly to *Trp53* and *Cdkn1a* KO. We have now included a clearer statement about this in page 6.

In contrast, *Zfp266* KO increases *Nanog*-GFP⁺ colony numbers without increasing *Nanog*-GFP⁻ colony numbers, suggesting that the reprogramming enhancement is not due to suppression of senescence and/or apoptosis.

Figure 2: Continuing on my foregoing question: The authors should provide a (possible) explanation for the abundance of mOrange⁺/Nanog⁻ cells in the two most efficient known reprogramming inhibitors TRP53 and Cdkn1a? Again, Zfp266 has a qualitatively different appearance in the alternative assays shown here. The authors should address this in the text and perhaps add data to convince that reprogramming is occurring at the same level/with the same dynamics as the previously described reprogramming inhibitors under Zfp266 KO conditions (see the point above). Again, FAM122a seems a very interesting target to follow up as well, given it's similarities in appearance to TRP53 and Cdkn1a, but it's not addressed as such in the text. Is there a reason for this?

As we stated above, *Trp53* and *Cdkn1a* KO increase number of *Nanog*-GFP⁺ colonies via suppression of reprogramming factor-induced senescence and apoptosis. The *Fam122a* KO phenotype indicates that *FAM122A* is involved in reprogramming factor-induced senescence and/or apoptosis, and it might be a molecule that links reprogramming factor expression and *Trp53* and *Cdkn1a* activation/induction. Thus, we agree with this reviewer that *Fam122a* is indeed an interesting gene, but in this manuscript we focused on *Zfp266* KO which facilitated iPSC generation independently from senescence/apoptosis suppression, which we found more interesting.

Figure 3: Why are the cells not followed up/analysed at later stages? The PCR plot shows that d7 cells during conversion are clearly different from MEFs, but they are still a long way from iPSCs (with fulfilled conversion). It would be important to show that the zfp266 KO cells can fully convert to iPSCs.

We appreciated and addressed this question in Supplementary Figures S4B-S4H. Please see the response to the reviewer 1's major point 3 above.

Figure 5; Line 221: the overlap between ATAC-seq peaks and DamID-seq was very limited, probably because the DamID-seq was performed in MEFs, and the ATAC-seq 72 hours into iPSC conversion. Although the data are quite convincing, it would be a nice addition to show Zfp266 binding sites at the 72 hour timepoint to explain this discrepancy and get more support for a direct link between the ATAC-seq data and actual Zfp266 binding

We have in fact performed Zfp266 DamID-seq 72 hours after Yamanaka factor expression. However, we could not obtain consistent data across technical replicates. Similarly, Oct4 DamID-seq at 72 hours after Yamanaka factor expression also gave us inconsistent results. We therefore concluded that DamID-seq does not work well during reprogramming probably due to extensive variability and heterogeneity of the cells. Antibodies against Zfp266 are not available, hence we tagged the endogenous coding sequence of Zfp266 with a Ty1-tag inserted just after the first methionine in ESCs for the purpose of ChIP-seq. While we confirmed the correct DNA sequence of the targeted allele, we could not detect Ty1-ZFP266 protein in ES cells either by Western blotting or immunofluorescence, thus gave up to make MEFs from the Ty1-tagged ZFP266 knock-in ESC line. While we could have performed Zfp266 ChIP-seq 72 hours after induction of exogenous tagged Zfp266 together with reprogramming factors, this would have required a large amount of starting reprogrammable MEFs, and the data would likely not represent genuine binding sites of endogenous Zfp266 because Zfp266 overexpression strongly inhibits iPSC generation. For these reasons, we hope this reviewer accepts the lack of Zfp266 binding site data at 72 hour time point, which have been complemented by other data.

Minor points

- it is unclear in the title, abstract and introduction in which species the analysis is done. There should be a clear mentioning of the species.

We have made it clear in the title, abstract and introduction now.

- The last paragraph of the intro reads much like an abstract and should be changed.

We appreciate this point and made it more concise.

- Have the authors considered making double or triple KOs of the different pluripotency blockers to see if there is an additive effect?

We took the reviewer's suggestion and performed double KO experiments. As shown in Supplementary Figures S3D, Zfp266 gRNA could enhance reprogramming in the presence of *Trp53*, *Cdkn2a*, or *Fam122a* gRNAs, all of which tend to increase the number of NANOG- partially reprogrammed colonies. Thus, it is likely that reprogramming enhancement by *Zfp266* KO is not due to circumvention of reprogramming factor-induced senescence and apoptosis. We described this results in page 7.

Supplementary Figures S3D. piggyBac-mediated reprogramming with double KO of roadblock genes. sgRNAs against *Zfp266* and other roadblock genes were cloned into rtTA and reprogramming factor expression vectors, respectively. Transposons with indicated sgRNA(s) were transfected into Cas9 expressing MEFs. 14 days later numbers of NANOG⁺ and NANOG⁻ colony were counted. Data represent an average of two independent experiments with 3 technical replicates, and error bars indicate SEM.

- BFP is not explained anywhere in the manuscript. I assume Blue Fluorescent Protein?

We apologize the lack of explanation. Yes, it is blue fluorescent protein, we added it now in page 13.

- In figure 3D, I feel it's better to indicate 'wt-zfp266' instead of 'sgRNA-resistant'

We apologize that the figure legend had an error. In the Figure 3D, all the samples had *Zfp266* sgRNA. In order to block *Zfp266* KO-mediated reprogramming enhancement, we needed to use sgRNA-resistant *Zfp266* (the second column from the left). All the KRAB mutants are made by modifying the KRAB domain of sgRNA-resistant *Zfp266*. We have corrected the error.

Reviewer #3 (Remarks to the Author):

This manuscript from Kaemena et al reports that the KRAB zinc finger protein (KRAB-ZFP) Zfp266 modulates somatic reprogramming through inhibiting the chromatin remodeling mediated by pioneering factors. They applied genome-wide CRISPR KO screening and identified Zfp266 as a novel repressor for somatic reprogramming. Knockout of Zfp266 improved reprogramming efficiency while overexpression (OE) of Zfp266 impedes reprogramming. By using DamID-seq and ATAC-seq, the authors observed that ZFP266 binds SINE-B1 elements adjacent to Yamanaka factors Oct4/Sox2/Klf4 binding sites and then suppresses chromatin opening mediated by these pioneer factors.

However, many of the data are over-interpreted, which not fully support their statements. For example, there is no direct evidence that SINE-B1 elements regulate the genes suggested by the authors, and it lacks function validations for those genes. This manuscript also lacks the biological insight in this field, thus I would suggest considering the manuscript more suitable for another journal.

We thank this reviewer for spending time on our manuscript. We answer to their concerns below one by one.

Major points

1. In Figure4, there are 15119 ZFP266 binding sites, but only 123 of which are MORs after *Zfp266*-KO, indicating that the major function of ZFP266 is still unknown instead of suppression chromatin accessibility as the authors claim.

We were also surprised by the small number of MORs in *Zfp266* KO MEFs. However, the loss of one suppressor does not necessarily result in chromatin opening in all loci. CpGs of B1 SINEs are also highly methylated in both ESCs and differentiated cells (Meisunner, *et al.*, Nature, 2008). In addition, we believe

activators need to be involved to observe significant effects on chromatin organisation. This could be likened to releasing the breaks of a car parked on a flat surface, doing so does not result in the car motion.

2. The Luciferase reporter assay cannot represent the effects of endogenous genomic elements faithfully. For Figure 6E, the authors claim that SINE act as regulatory elements for the *B3gnt3*, *Piwi2*, *Snx20*. It is required to delete those SINE-loci and to check if such deletions impair the expression of respective gene. A SINE-B1 element is very short, allowing easy validation by CRISPR-KO.

We agree that deleting SINE from the endogenous *B3gnt3*, *Piwi2*, *Snx20* loci and monitoring expression of those genes during reprogramming would have been better approach. However, it is not trivial to validate that these SINEs act as regulatory elements with CRISPR-KO. It was impossible to design good gRNAs to remove those SINEs without disturbing the KLF4 binding motif, unlike modifying reporter plasmids. All the B1 SINEs in the *B3gnt3*, *Piwi2* and *Snx20* regulatory elements have a KLF4 binding motif at the beginning of SINE as predicted from the new Figure 5K, and we believe those KLF4 motifs are important for the up-regulation of those genes by reprogramming factors. Designing gRNAs within the SINE itself is also not ideal as SINEs are widespread in the genome. Therefore, we used the luciferase reporter assay as a reasonable alternative.

We agree that although luciferase reporter assays are widely used to study the regulatory control of a gene of interest, they do not always reflect the effects of endogenous genomic elements faithfully, therefore we added a sentence in page 14 to highlight this caveat.

3. Figure 6E, the ChIP-seq data signal seems too sparse to be real binding signal, especially at the *B3gnt3* loci. Input control is required to exclude any biased interpretation.

All the ChIP-seq tracks and heatmaps presented in this work show are input control subtracted data. We added this information in the methods and the figure legends when relevant. We do agree that those ChIP-seq signals are not as strong as in pluripotency gene loci in ESCs/iPSCs. However, we decide to show them anyway to provide information on how OSKM binding looks at that stage of reprogramming.

4. The mechanism of *Zfp266*-KO facilitates reprogramming is still unknown. The authors have listed a few genes, but whose function seems still unclear. Can overexpression of one or more these genes promote reprogramming?

We do not insist that B3gnt3, Piwil2, and/or Snx20 affects reprogramming efficiency, although Piwil2 is associated with stem cell self-renewal and is expressed in mouse ESCs. We chose those gene loci purely to investigate gene expression regulation by ZFP266 and reprogramming factors, as 1) they are significantly up-regulated in *Zfp266* KO reprogramming, 2) have more open region(s) with SINE and KLF4 binding at early time point of reprogramming. However, we do believe the mechanism proposed in the Figure 60 supported by many data in this manuscript is important for many pluripotency gene up-regulation.

Figures 60. Mechanistic model of how *Zfp266* KO enhances reprogramming. B1 SINEs are strongly associated with nucleosome and often have KLF and SOX motif at the head and tail due to G and A rich sequence, respectively. ZFP266 binds to some of SINEs in a cell type- and context-dependent manner. Upon reprogramming factor expression, ZFP266 recruited to B1 SINE with KLF/SOX motifs at its head or tail, respectively, impedes chromatin opening (top). *Zfp266* KO results in increased chromatin accessibility in those loci, facilitating pluripotency gene expression.

In order to identify pluripotency genes whose up-regulation can potentially be enhanced in *Zfp266* KO reprogramming through this mechanisms, we looked for iPSC's open chromatin loci with B1 SINE. Of the 178,112 iPSC ATAC-seq peaks, 22% (39,353) had B1 SINEs, which were mostly closed in MEFs (Figure 7A). ~70 bp from the midpoint (i.e. head and tail regions) of these SINEs are enriched in KLF and SOX motifs (Figure 7B), resulting in 21, 150 iPSC ATAC-seq peaks with at least one SINEs and one KLF/SOX motif. 76 out of the 137 genes with a GO term "stem cell population maintenance" were associated with such ATAC-seq peaks, many of which are significantly upregulated in reprogramming following *Zfp266* depletion, including very important genes for pluripotency such as *Nanog*, *Esrrb* and *Dppa2* (Figures 7C and 7D). Up-regulation of these pluripotency genes during reprogramming could be facilitated by lack of B1 SINE/ZFP266-mediated chromatin closing in *Zfp266* KO cells.

Supplementary Figures S7. B1 SINEs in iPSC open chromatin regions. **A.** Of 178,112 iPSC ATAC-peaks, 22% (39,353) has at least one B1 SINE, majority of which are closed in MEFs. **B.** Those B1 SINEs at iPSC ATAC-peaks have KLF4 and SOX3 motif enrichment at the head and the tail of SINE. **C.** Gene associated with 12,150 iPSC ATAC-seq peaks with at least one B1 SINE and KLF/SOX motif are enriched in genes with a GO term “stem cell population maintenance”. Well known pluripotency associated genes are highlighted in magenta, up-regulation of many of which were accelerated in reprogramming of *Zfp266* KO MEFs. **D.** Pluripotency gene loci with iPSC ATAC-seq peaks (red), B1 SINE (green) and KLF4 ChIP-seq signals (Blue). B1 SINE with KLF4 motif is indicated in dark green, and proximity of B1 SINEs to KLF4 ChIP-seq peaks is indicated in red arrows. Input subtracted ChIP-seq data are shown.

Reviewer #1 (Remarks to the Author):

The authors have address my comments and concerns in full and I am supporting publications of the revised manuscript. The new analysis in Figure is vey interesting The only remaining suggestion I would have is to consider replacing the term 'pioneering factors' in the title (i.e. with reprogramming factors or pluripotency factors) as all transcription factors mich be pioneering factors in one way or the other.

Reviewer #2 (Remarks to the Author):

I want to thank the authors for their elaborate answers to the points that were raised and appreciate the additions that were made in the text and supplemental figures, and the explanations that were provided for their inability to provide some data which I felt would be a good contribution to the paper (such as the response to my comments on Figure 5).

As the authors propose in their rebuttal, I do suggest to include the human data on ZNF571 and Alus presented in the response figure 3 as a supplemental figure with a brief description of their search for the human functional ortholog of mouse Zfp266, because this would be of great importance for researchers who want to start extrapolating the findings in this study to human iPSC reprogramming.

Finally, the title is still unclear about the species the study is about, and because the findings cannot be directly translated to human iPSC reprogramming, this seems quite important to me. I suggest to add 'mouse iPSC' before 'reprogramming' in the title.

With these final 2 suggestions, which I leave for the editor to decide, I now find the study suitable for publication in Nature Communications.

Frank Jacobs

Reviewer #3 (Remarks to the Author):

I appreciate the authors' efforts in clarifying my queries. Overall, I feel the study is improved.

However, for the first round question2, is none less of Klf4. Knock-out of those SINE-B1 loci with two sgRNA flanked the repeat regions, and then check if such deletions impair the expression of respective gene, as performed by two similar studies^{1,2}. In my view, this is the only direct evidence to support the SINE-B1 elements regulating those genes expression as the authors claimed. Alternatively, mutation of the ZFP266 motif and KLF4 motif control will be helpful in addressing the potential side effects of luciferase reporter assays.

Reviewer #1 (Remarks to the Author):

The authors have address my comments and concerns in full and I am supporting publications of the revised manuscript. The new analysis in Figure is vey interesting The only remaining suggestion I would have is to consider replacing the term 'pioneering factors' in the title (i.e. with reprogramming factors or pluripotency factors) as all transcription factors mich be pioneering factors in one way or the other.

We appreciate this reviewer's positive comments. We have changed the title accordingly.

Reviewer #2 (Remarks to the Author):

I want to thank the authors for their elaborate answers to the points that were raised and appreciate the additions that were made in the text and supplemental figures, and the explanations that were provided for their inability to provide some data which I felt would be a good contribution to the paper (such as the response to my comments on Figure 5).

As the authors propose in their rebuttal, I do suggest to include the human data on ZNF571 and Alus presented in the response figure 3 as a supplemental figure with a brief description of their search for the human functional ortholog of mouse Zfp266, because this would be of great importance for researchers who want to start extrapolating the findings in this study to human iPSC reprogramming.

Finally, the title is still unclear about the species the study is about, and because the findings cannot be directly translated to human iPSC reprogramming, this seems quite important to me. I suggest to add 'mouse iPSC' before 'reprogramming' in the title.

With these final 2 suggestions, which I leave for the editor to decide, I now find the study suitable for publication in Nature Communications.

Frank Jacobs

We appreciate this reviewer's positive comments and constructive suggestions. We have integrated the human data as Supplementary Figure S8 and changed the title accordingly.

Reviewer #3 (Remarks to the Author):

I appreciate the authors' efforts in clarifying my queries. Overall, I feel the study is improved.

However, for the first round question2, is none less of Klf4. Knock-out of those SINE-B1 loci with two sgRNA flanked the repeat regions, and then check if such deletions impair the expression of respective gene, as performed by two similar studies^{1,2}. In my view, this is the only direct evidence to support the SINE-B1 elements regulating those genes expression as the authors claimed. Alternatively, mutation of the ZFP266 motif and KLF4 motif control will be helpful in addressing the potential side effects of luciferase reporter assays.

We thank this reviewer for understanding the difficulty of removing only B1 SINE from the genome leaving the adjacent KLF4 motif intact.

We also appreciate the suggestion by this reviewer to make a mutation in ZFP266 and KLF4 binding motif. However, while we now know that ZFP266 binds to B1 SINE, we still do not know the actual binding motif within B1 SINE. A ZFP binding motif prediction tool (the PWM predictor; <http://zf.princeton.edu/pwmBatch.php>) also did not identify a sequence that matches with the B1 SINE consensus sequence, as we wrote in response to reviewer 1 minor point 1 previously. This is why we deleted the entire B1 SINE sequence from the MOR reporters (Figure 6H-L).

It would be possible to remove KLF4 motifs from the B1 SINE deleted reporters, and see if OSKM-mediated reporter activity induction is diminished. However, in the *Snx20* MOR reporter used in Figure 6L, there are x15 KLF4 motifs, as well as x1 OCT4 motif and x10 SOX2 motifs, without considering a single nucleotide mismatch. If we permit a single nucleotide mismatch, there are x84 KLF4 motifs (as well as x31 OCT4 motifs and x127 SOX2 motifs), and we do not know which motif(s) is critical for the OSKM-mediated reporter induction. We could mutate all, but we do not think changing so many nucleotides is a much better control than the left side of Figure 6L, which demonstrated that the *Snx20* reporter activity in MEFs did not change by the deletion of B1 SINE when there was no OSKM induction.

SOX2 motif

BYWTTNT

OCT4 motif

AWATGC, AAATAC, ATGCAHAT

KLF4 motif

GGGHGK

Y = C or T, N = A or C or G or T, W = A or T, H = A or C or T, K = G or T (Motifs from Soufi, et al, Cell, 2015)

Response Figure R1. OCT4, SOX2, KLF4 motifs in the *Snx20* MOR reporter. Top: DNA sequence used in the reporter. Blue = B1 SINE, yellow = MOR, green = OCT4 motif, pink = SOX2 motif, purple = KLF4 motif. The marked motifs were determined based on the canonical motifs and motifs identified by OCT4, SOX2, KLF4 ChIP-seq at 48 hours of reprogramming by Soufi, et al, Cell, 2015)

We hope the reviewer agrees that our data sufficiently support our model that ZFP266 suppresses OSKM-mediated gene up-regulation via binding to B1 SINE without the additional reporter assay.